# Integrating Relation Dependencies and Textual Semantics for Coherent Logical Reasoning over Temporal Knowledge Graphs

## Abstract

Temporal knowledge graphs (TKGs) reflect the evolution patterns of facts, which can be summarized as logical rules and applied to forecast future facts. However, existing logical reasoning methods on TKGs face two limitations: 1) A lack of efficient strategies for extracting logical paths. 2) Insufficient utilization of structural and textual information. To bridge these gaps, we propose CoLR, a two-stage framework that mines relation dependencies and textual semantics for **Co**herent **L**ogical **R**easoning over TKGs. In the first stage, we construct a temporal relation structure graph (TRSG) composed of relations and cohesion weights between them. Besides, we define a novel time-fusion search graph (TFSG) along with TRSG to facilitate efficient and reliable temporal path searching. In the second stage, the textual content and timestamp sequences from these paths undergo encoding via a pre-trained language model and a time sequence encoder to accurately capture potential logical rules. Additionally, for quadruplets missing paths, historical edges sampled based on relation cohesion are used as supplements. Given the limitations of existing benchmark datasets in evaluating accuracy, generalization, and robustness, we construct three new datasets tailored to transductive, inductive, and few-shot scenarios, respectively. These datasets, combined with four widely-used real-world datasets, are employed to evaluate our model comprehensively. Experimental results demonstrate that our approach significantly outperforms existing methods across all three scenarios. Our code is available at https://anonymous.4open.science/r/CoLR-0839.

## 1 Introduction

Temporal Knowledge Graphs (TKGs) are a pivotal method for representing the dynamic facts of the real world. Due to their high application value in personalized recommendations (Wang et al., 2022) and conversational systems Shang et al. (2022), they have attracted widespread research attention. Reasoning tasks on TKGs encompass not only filling missing links within historical subgraphs (interpolation reasoning) but also predicting future interactions among entities by analyzing historical patterns (extrapolation reasoning). However, to achieve extrapolation reasoning, inductive capabilities are significant, as new entities emerge over time. As shown in Figure 1, once *Donald Trump* assumes the presidency, he is regarded as a new entity. Therefore, more fundamental connections between facts should be uncovered, such as the logical correlations between relations.

Recently, several TKG Reasoning (TKGR) methods have been proposed, where multi-hop logical reasoning methods learn logical rules from the historical multi-hop relation paths Liu et al. (2022); Niu & Li (2023); Xiong et al. (2023); Mei et al. (2024). During the reasoning process, these methods predict target entities by applying these rules to give confidence scores to each candidate. Assuming the stability of these logical rules, such methods can effectively generalize to inductive scenarios, with the logical rules offering interpretability for the reasoning process. For example, *Imran Khan* in Figure 1(a) can be found as logical correlation between relations *Express intent to meet or negotiate* and *Make a visit* has been learned in Figure 1(b)

The core of multi-hop logical reasoning lies in finding reliable multi-hop paths and their scoring strategies. Traditional symbolic methods Omran et al. (2019); Liu et al. (2022) perform extensive

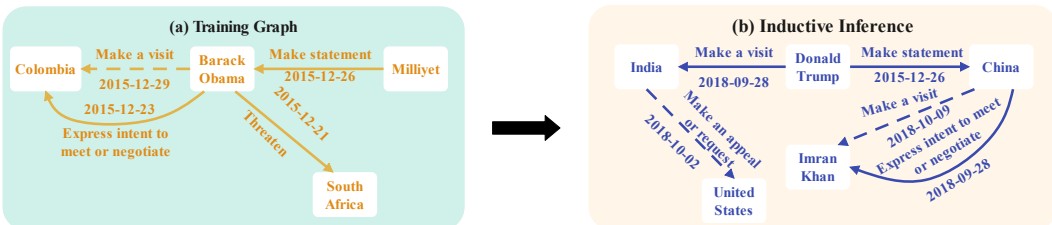

Figure 1: Inductive inference over TKGs: training with historical data and inferring the future facts involving new entities, where the quadruplets to be predicted are connected by dashed lines.

random walks to sample relation paths and assess rule confidence with global frequency information. However, this process may incur substantial computational overhead and potentially overlook low-frequency rules. Recent methods Xiong et al. (2023); Mei et al. (2024) simplify the rule learning process by encoding relation paths and evaluating confidence based on the semantic similarity between paths. Nonetheless, the majority of paths between the subject and object are predominantly irrelevant noise paths, which may potentially interfere with inference. Moreover, existing methods exhibit significant limitations when confronted with quadruplets lacking direct connectivity between subject and object entities. Furthermore, these methods insufficiently utilize the rich structural and textual information in temporal graphs. Symbolic methods leverage the frequency information of relations, while neural methods focus on graph structural information. Both overlook the positive role of the structural dependencies between relations and textual semantics in logical reasoning.

To address these challenges, we propose a novel **Co**herent **L**ogical **R**easoning method, named CoLR, which is a two-stage portable framework designed to handle inductive scenarios. In the first stage, we introduce a temporal relation structure graph (TRSG), where nodes represent relations, and weighted edges reflect the cohesion between relations. The main idea comes from the concept of sentence cohesion in linguistics Halliday & Hasan (1976), where words in a coherent sentence should have semantic associations. Similarly, effective relation paths should also demonstrate close connections between their elements. We design a time-fusion search graph (TFSG) and develop a novel search algorithm to guide path selection along with cohesion weights between relations, thus efficiently extracting reliable paths. If no connected path exists, we sample historical edges based on relation cohesion and temporal dependency as information supplements. In the second stage, we extract textual sequences and timestamp sequences of paths, and introduce a pre-trained language model (PLM) and time sequence encoder for temporal and logical semantic encoding. The textual semantics embedded in textual descriptions helps reveal the logical correlations between relations, such as "Express intent to meet or negotiate" and "make a visit" in Figure 1. During the reasoning process, we combine the embeddings of text and timestamps and evaluate the confidence of quadruplets based on semantic similarity.

The proposed CoLR framework demonstrates significant transferability to previous logical reasoning methods, effectively optimizing their learning efficiency and reasoning performance. Notably, we introduce a novel confidence evaluation function for discrete logical reasoning methods, predicated on the concept of cohesion. This innovative approach circumvents the substantial redundant computations inherent in conventional methods while maintaining accuracy.

To comprehensively evaluate the efficacy of our methodology, we construct three novel datasets that address the deficiencies in existing benchmarks, particularly in assessing accuracy, generalization, and robustness. We conduct extensive experimentation across seven distinct datasets, encompassing three pivotal experimental paradigms: transductive, inductive, and few-shot learning scenarios. Experimental results show that our proposed method consistently achieves state-of-the-art (SOTA) results under the three settings. Comprehensive ablation experiments and case studies verify the effectiveness of the components we proposed in capturing the logical correlations between relations and enhancing performance.

## 2 RELATED WORK

**Representation-based methods for TKGR.** Temporal Knowledge Graph Embedding (TKGE) is a classic representation-based method for TKGR tasks. These methods project the TKG into

various complex embedding spaces, such as quaternion and hyperbolic manifold spaces, to learn vector representations of entities, relations, and timestamps. They have achieved significant success in interpolation settings, exemplified by methods like TTransE Leblay & Chekol (2018), TA-DisMult García-Durán et al. (2018), TNTComplEx Lacroix et al. (2020), BoxE Abboud et al. (2020), TeLM Xu et al. (2021), HGE Pan et al. (2024). However, due to a lack of scalability, these methods struggle to adapt to extrapolation reasoning. Recent studies have utilized RGCN Schlichtkrull et al. (2018) to aggregate entity representations across historical snapshots and evolve these representations over time using RNNs to address future time predictions. Despite their advancements, these methods still heavily rely on entity representations, which makes them less suitable for inductive scenario He et al. (2021); Jin et al. (2020b); Li et al. (2021; 2022b;a); Liang et al. (2023); Zhang et al. (2023a;b); Li et al. (2024). Additionally, the black-box nature of complex neural networks makes their predictive outcomes difficult to interpret intuitively.

**Rule-based methods for TKGR.**  Rule-based methods learn first-order logical rules from the graph and apply these rules to infer missing links. These methods also achieve stable predictions in inductive scenarios, as demonstrated by AMIE+ Galárraga et al. (2015), DRUM Sadeghian et al. (2019) and RNNLogic Qu et al. (2021). In TKGs, timestamps mark the sequence of events, making the logical associations within them easier to uncover. Therefore, symbolic methods like StreamLearnerOmran et al. (2019), TLogic Liu et al. (2022) and LCGE Niu & Li (2023) attempt to sample historical multi-hop paths and extract reliable temporal logic rules from them. These rules, which do not change with time or entities, are naturally suited for inductive prediction of future events and provide interpretable reasoning. However, these methods require extensive random walks and corresponding rule evaluations, leading to high computational costs. Recent approaches have introduced GRUs to encode paths specific to query quadruplets and assess the credibility of rules based on semantic similarity, thus simplifying rule learning and speeding up the reasoning process. This strategy, which integrates neural methods, allows for more flexible and comprehensive learning of logical rules, achieving superior performance in TKGR tasks. However, these methods Mei et al. (2022); Xiong et al. (2023); Mei et al. (2024) still struggle with the absence of connected paths from subject to object. Moreover, they overlook the positive influence of the logical semantics embedded in textual information on the learning of logical rules.

## 3 PRELIMINARIES

**Temporal Knowledge Graph(TKG).**  A TKG $\mathcal{G}$ is a complex structure used to record known facts, where $\mathcal{G} \subseteq \mathcal{E} \times \mathcal{R} \times \mathcal{E} \times \mathcal{T}$, $\mathcal{E}$ represents the set of entities, $\mathcal{R}$ represents the set of relations, and $\mathcal{T}$ is the set of timestamps. At a specific point in time, all facts make up a temporal subgraph $\mathcal{G}_i = \{(s_{ij}, r_{ij}, o_{ij}, t_i)\}$, where $s_{ij}, o_{ij} \in \mathcal{E}$ represent the subject and object of the $j$th event in the subgraph, respectively, $r_{ij} \in \mathcal{R}$ represents the relation, and $t_i \in \mathcal{T}$ is the $i$-th timestamp. The entire TKG can be divided along timestamps into a series of consecutive temporal subgraphs: $\mathcal{G} = \{\mathcal{G}_i\}_{1 \leq i \leq |\mathcal{T}|}$. To enhance the connectivity of the graph, each relation $r \in \mathcal{R}$ is supplemented with an inverse relation $r^-$, and the corresponding inverse edges are denoted as $(o, r^-, s, t)$.

**Link Forecasting.**  Link forecasting is one of the core tasks of TKGR. This task involves a historical graph $\mathcal{G}$ containing known facts, with the maximum timestamp $\hat{t}$, and aims to predict future possible links. Specifically, for a query quadruplet in the form $(s, r_q, ?, t_q)$, where $t_q > \hat{t}$, an ordered list of candidate entities needs to be generated. During the training process, the goal is to increase the score of the quadruplets comprising the correct candidate entities and decrease the score of those with incorrect candidates. For cases where the subject is being predicted, the query quadruplet is formalized as $(o, r_q^-, ?, t_q)$.

**Temporal Logical Rule.**  A first-order logical rule is composed of a rule head and several binary rule body atoms connected in sequence, and can be formalized in a TKG as a first-order temporal logical rule of length $l$:

$$\Upsilon_l : P_{l+1}(x, y, t_{l+1}) \leftarrow P_1(x, z_1, t_1) \wedge P_2(z_1, z_2, t_2), ..., \wedge P_l(z_{l-1}, y, t_l), \tag{1}$$

where $x$ represents the subject entity, $y$ represents the object entity, and $z_i$ can be any entity in the graph. Each $P_i$ represents a predicate, typically instantiated as a relation. $t_1, t_2, \ldots, t_{l+1}$ denote the timestamps for each atom and must satisfy a non-decreasing constraint: $t_1 \leq t_2 \leq \ldots \leq t_l < t_{l+1}$.

The left side of $\Upsilon_l$ is the rule head $H_\Upsilon$, and the right side is the rule body $B_\Upsilon$. If $B_\Upsilon$ is true and satisfies the temporal constraints, then its $H_\Upsilon$ is inferred to be true. In a TKG, if $B_\Upsilon$ can be grounded,

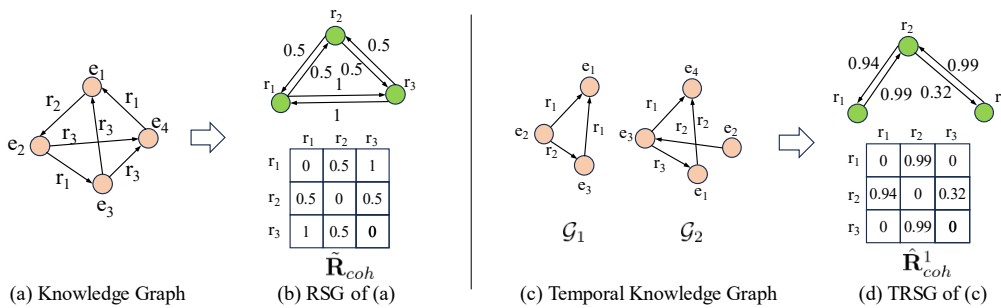

Figure 2: A simple example of an RSG and a TRSG, where yellow nodes and green nodes represent entities and relations, respectively. The inverse and self-loop edges are omitted for brevity.

meaning all variables can be replaced with entities, relations, and timestamps from the TKG, it is considered to be true. The grounded $B_\Upsilon$ represents a multi-hop temporal path within the TKG, accordingly, $H_\Upsilon$ can be grounded as a quadruplet.

## 4 CONSTRUCTING TEMPORAL RELATION STRUCTURE GRAPH

### 4.1 RELATION STRUCTURE GRAPH

To capture the structural dependencies between relations, we define the positional patterns of relations as follows:

**Definition 1.** *Given a directed knowledge graph $\mathcal{G}$, a node $e$, and its connected edge set $\mathcal{F}$, the positional pattern between any two relations $r_1$ and $r_2$ in $\mathcal{F}$ ($r_1 = r_2$ is possible) is:*
*(i) Coherent: $e$ serves as the subject for $r_1$ and the object for $r_2$, or as the object for $r_1$ and the subject for $r_2$;*
*(ii) Homologous: $e$ serves as the subject for both $r_1$ and $r_2$;*
*(iii) Converging: $e$ serves as the object for both $r_1$ and $r_2$.*

If two relations are always connected to the same entity and exhibit the same positional pattern, then there is a definite structural dependency between them. We define structural dependencies based on positional patterns as *cohesion*, *homology*, and *convergence*. Some recent research endeavors Lee et al. (2023); Chen et al. (2021) have explored the optimization of graph neural networks by leveraging the homogeneity and homology between relations. Our approach, however, diverges significantly from these efforts. As a logical reasoning method, our primary focus is on the principle of cohesion. We estimate cohesion weights by calculating the frequency of two relations occurring in the coherent positional pattern.

For a static knowledge graph $\mathcal{G} = (\mathcal{E}, \mathcal{R}, \mathcal{F})$, we define two entity-relation matrices $\mathbf{E}_s \in \mathbb{R}^{|\mathcal{E}| \times 2|\mathcal{R}|}$ and $\mathbf{E}_o \in \mathbb{R}^{|\mathcal{E}| \times 2|\mathcal{R}|}$, where the subscripts "s" and "o" represent subject and object, respectively. $\mathbf{E}_s[i, j]$ denotes the element at the $i$-th row and $j$-th column, recording the number of times the entity $e_i$ in the graph is connected as a subject to the relation $r_j$ (including the inverse relation). Similarly, $\mathbf{E}_o[i, j]$ records the number of times $e_i$ is connected as an object to the relation $r_j$. Considering the variance in interaction frequencies of entities within the graph, we introduce a degree diagonal matrix for entities to compute the frequency of each entity's interaction with relations, given by $\tilde{\mathbf{E}}_s = \mathbf{D}_s^{-1} \mathbf{E}_s$ and $\tilde{\mathbf{E}}_o = \mathbf{D}_o^{-1} \mathbf{E}_o$. Subsequently, a cohesion matrix is defined as: $\tilde{\mathbf{R}}_{coh} = \tilde{\mathbf{E}}_o^T \tilde{\mathbf{E}}_s$.

A coherent relation structure graph (RSG) can be constructed through $\tilde{\mathbf{R}}_{coh}$. As shown in Figure 2(a), $\tilde{\mathbf{R}}_{coh}[i, j]$ represents the weight of the edge between nodes i and j in the graph. In multi-hop logical reasoning, a multi-hop relation path should be coherent, thus the RSG can quickly assess the confidence of a rule based on cohesion between relations. However, the method for constructing RSG in static graphs cannot be directly applied to TKGs. Because TKGs consist of multiple subgraph sequences, the structural dependencies between relations exist not only within the same subgraph but also across different subgraphs.

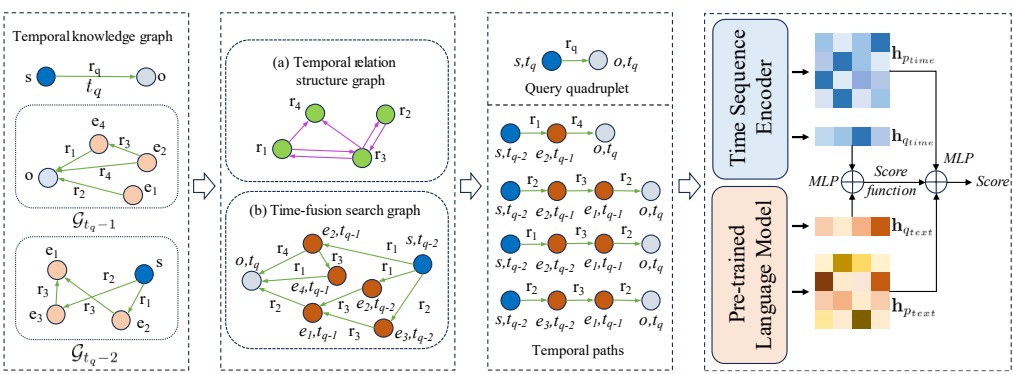

Figure 3: Overview of CoLR framework. The green and purple lines represent the relations and the cohesion weights between them, respectively.

## 4.2 TEMPORAL RELATION STRUCTURE GRAPH

In a TKG, the closer the time, the stronger the association between facts. Thus for link forecasting tasks, reasoning is often based on the most recent facts Liu et al. (2022); Li et al. (2022a); Mei et al. (2022). To align with the reasoning phase, we define a time window $\omega$ and focus on the relation dependencies within $\omega$ subgraphs. Theorem 1 demonstrates the calculation method for the coherent matrix with $\omega$ time windows $\mathbf{R}_{coh}^{\omega}$.

**Theorem 1.** *Let $\mathcal{G} = \{\mathcal{G}_i\}_{1 \le i \le |\mathcal{T}|}$ be a TKG, $\mathbf{E}_s^i$ be the subject-relation matrix of the $i$-th subgraph$\mathcal{G}_i$, $\mathbf{E}_o^i$ be the corresponding object-relation matrix, $0 < \omega \le |\mathcal{T}|$ be a time window. A cohesion relation structure matrix $\mathbf{R}_{coh}^{\omega}$ without normalization can be calculated as follows:*

$$\mathbf{R}_{coh}^{\omega} = \sum_{j=\omega}^{|\mathcal{T}|} (\sum_{i=j-\omega+1}^{j} \mathbf{E}_o^{i^T}) \mathbf{E}_s^j + \sum_{j=1}^{\omega-1} (\sum_{i=1}^{j} \mathbf{E}_o^{i^T}) \mathbf{E}_s^j. \tag{2}$$

We illustrate formulation (2) with the following simple example. For a detailed proof, refer to Appendix B.1.

**Example 1.** *Assume $\mathcal{G} = \{\mathcal{G}_1, \mathcal{G}_2, \mathcal{G}_3\}$, $\omega = 2$, the coherent matrix can be calculated by*

$$\mathbf{R}_{coh}^2 = (\mathbf{E}_o^{1^T} + \mathbf{E}_o^{2^T})\mathbf{E}_s^2 + (\mathbf{E}_o^{2^T} + \mathbf{E}_o^{3^T})\mathbf{E}_s^3 + \mathbf{E}_o^{1^T}\mathbf{E}_s^1$$
$$= \sum_{j=2}^{3} (\sum_{i=j-1}^{j} \mathbf{E}_o^{i^T})\mathbf{E}_s^j + \sum_{j=1}^{1} (\sum_{i=1}^{j} \mathbf{E}_o^{i^T})\mathbf{E}_s^j.$$

Following Section 4.1, we introduce degree matrices $\mathbf{D}_s^i$ and $\mathbf{D}_o^i$ to normalize the matrices of each subgraph $\tilde{\mathbf{E}}_s^i = \mathbf{D}_s^i \mathbf{E}_s^i$ and $\tilde{\mathbf{E}}_o^i = \mathbf{D}_o^i \mathbf{E}_o^i$, and replace them in formulation (2) to obtain the matrix $\tilde{\mathbf{R}}_{coh}^{\omega}$. We normalize the matrix $\tilde{\mathbf{R}}_{coh}^{\omega}$ to obtain the final temporal cohesion matrix:

$$\hat{\mathbf{R}}_{coh}^{\omega} = \frac{\tilde{\mathbf{R}}_{coh}^{\omega}}{\|\tilde{\mathbf{R}}_{coh}^{\omega}\|_2 + \delta}, \tag{3}$$

where $\delta \in (0, 0.1)$ is a parameter introduced to prevent the denominator from being zero. Similar to Section 4.1, a TRSG can be constructed based on $\hat{\mathbf{R}}_{coh}^{\omega}$, as illustrated in Figure 2(b). We provide an analysis of the computational complexity of TRSG construction in Appendix A.3.

## 5 COHERENT LOGICAL REASONING OVER TKGS

### 5.1 TIME-FUSION PATH EXTRACTING

As illustrated in Figure 3, in the first stage, our goal is to extract temporal logical paths corresponding to query quadruplets. To achieve this, we have designed a time-fusion algorithm. Specifically,

for a query quadruplet $(s, r_q, o, t_q)$, we retrieve the multi-hop neighbors of $s$ and $o$ from historical subgraphs and construct a time-fusion search graph (TFSG) $\mathcal{G}_{tf}$ using their intersections, as illustrated in Figure 3(b). Each entity in $\mathcal{G}_{tf}$ is annotated by a timestamp, denoted as $e_i^t = [e_i, t]$. Correspondingly, a $l$-hop path $(e_1^{t_1}, r_1, e_2^{t_2}, ..., e_{l-1}^{t_l}, r_l, e^{t_{l+1}})$ in $\mathcal{G}_{tf}$ can be converted to a unique temporal path $\{(e_1, r_1, e_2, t_1), ..., (e_l, r_l, e_{l+1}, t_l)\}$.

The path searching starts from $o^{t_j}$ until reaching $s^{t_i}$, where $t_i \leq t_j$. The selection of the next edge is jointly determined by the time interval and the relation cohesion. For current entity $e_j^{t_j}$, probability of next edge over time is given by

$$P_{time} = \frac{exp(t_n - t_j)}{\sum_{(e_i^{t_i}, r_i, e_j^{t_j}) \in \mathcal{N}_j} exp(t_i - t_j)}, \tag{4}$$

where $\mathcal{N}_j$ is the neighbors of $e_j^{t_j}$, $t_n$ is the timestamp of next edge. Similarly, the probability of cohesion if given by

$$P_{coh} = \frac{exp(\hat{\mathbf{R}}_{coh}^{w^T}[r_j, r_n])}{\sum_{(e_i^{t_i}, r_i, e_j^{t_j}) \in \mathcal{N}_j} exp(\hat{\mathbf{R}}_{coh}^{w^T}[r_j, r_i])}, \tag{5}$$

where $\hat{\mathbf{R}}_{coh}^{\omega^T}[r_j, r_i]$ represents the cohesion when $r_j$ is next to $r_i$ in a temporal path. We combine $P_{time}$ and $P_{coh}$ to get the score for the next edge of $e_j^{t_j}$: $P_{next} = P_{time} + P_{coh}$.

A path with $l$ hop $(o^{t_j}, r_1, e_1^1, ..., e_{l-1}^{l-1}, r_l, s^{t_i})$ is reversed to $(s^{t_i}, r_l, e_{l-1}^{l-1}, ..., e_1^1, r_1, o^{t_j})$, forming a temporal path. To ensure the reliability of the extracted paths, we limit the path length to $L$ and the number of paths to $K$. If no path is found, we employ path supplement strategy (PSS) to collect historical contextual descriptions for these quadruplets. Specifically, we utilize $P_{next}$ to sample an edge from the historical neighbor edges of either $s$ and $o$ as a historical supplementary path. The sampled path offers crucial contextual descriptions for $(s, r_q, o, t_q)$, aiding in determining its significance. The detailed algorithm implementation are shown in Appendix A.1.

## 5.2 Joint Encoding of Time and Text Sequence

Given a temporal path $\{(s, r_1, e_1, t_1), ..., (e_{l-1}, r_l, o, t_l)\}$, we separate the timestamps to obtain an entity-relation path $(s, r_1, e_1, ..., r_l, o)$ and a timestamp sequence $(t_1, t_2, ..., t_l)$.

**Text Sequence Encoding.** The text sequence encompasses a textual description of entities and relations in $(s, r_1, e_1, ..., r_l, o)$. In contrast to previous approaches, we incorporate the entire path, not just the relation path, to generate text representations. We utilize ";" to delimit the text of each entity or relation, with "[SEP]" marking the end of a sentence. In the case of inverse relation $r^-$, we reverse the textual order of $r$ to generate the corresponding text. Then we introduce a PLM to encode the text sequence:

$$\mathbf{h}_{text} = \mathcal{M}(\Phi([\Gamma(s); \Gamma(r_1); ...; \Gamma(r_l); \Gamma(o)])), \tag{6}$$

where $\Phi(\cdot)$ denotes any PLM encoder, $\mathcal{M}$ is meanpooling operation, $\Gamma(\cdot)$ represents the textual description of an entity or a relation. $\mathbf{h}_{text} \in \mathbb{R}^d$ is the textual representation of the temporal path.

**Time Sequence Encoding.** We compute the difference between each element in $(t_1, t_2, ..., t_l)$ and $t_q$ to create a time sequence $(t_q - t_1, t_q - t_2, ..., t_q - t_l)$. The time sequence reflects the temporal feature associated with logical rules. For instance, occurrences of "make a visit" and "have a visit" tend to coincide. When they are distant in time, they are likely to be logically unrelated. We leverage Time2Vec Kazemi et al. (2019) and GRU to encode the time sequence:

$$\phi(t) = \sqrt{\frac{1}{d}} \left[ \cos(\mathbf{w}_1 t + \mathbf{p}_1), \cdots, \cos(\mathbf{w}_d t + \mathbf{p}_d) \right],$$
$$\mathbf{h}_{time} = GRU([\phi(t_q - t_1), \phi(t_q - t_2), ..., \phi(t_q - t_l)]), \tag{7}$$

where $\mathbf{w}, \mathbf{p} \in \mathbb{R}^d$ are learnable parameter vectors, $\mathbf{h}_{time} \in \mathbb{R}$ is the temporal representation of the temporal path. Finally, a MLP is used to combine $\mathbf{h}_{text}$ and $\mathbf{h}_{time}$ to generate the final representation: $\mathbf{h}_p = MLP([\mathbf{h}_{text}; \mathbf{h}_{time}])$. The two-stage framework of CoLR ensures high scalability, which is discussed in detail in Appendix A.2.

## 5.3 TRAINING REGIME

Consistent with prior research Mei et al. (2022); Su et al. (2023), we employ semantic similarity to evaluate the confidence of each temporal path. Then we score the query quadruplet with the highest confidence among all temporal paths, normalized as follows:

$$score(q) = \max_{p_i \in \mathcal{P}}\{cosine(\mathbf{h}_q, \mathbf{h}_{p_i})\}, \tag{8}$$

where $cosine(\cdot, \cdot)$ is cosine similarity function, $\mathbf{h}_{p_i}$ denotes the embedding of $i$-th path in extracted temporal paths set $\mathcal{P}$. To train our model, we treat the facts in the TKG as positive quadruplets and generate N corresponding negative quadruplets by replacing the subject or object. During training, we leverage cosine embedding loss to optimize our model, which is formulated as follows:

$$\mathcal{L} = \begin{cases} 1 - score(q) & y = 1, \\ max(0, score(q) - \gamma) & y = -1, \end{cases} \tag{9}$$

where $\gamma$ is the margin of confidence, $y \in \{-1, 1\}$ denotes the label of a quadruplet.

## 6 EXPERIMENTAL RESULTS

### 6.1 EXPERIMENT SETUP

**Datset & Evaluation.** To validate the performance of our model in a transductive setting, we initially selected four public datasets for comparative experiments on link forecasting task. These datasets include ICEWS14 García-Durán et al. (2018), ICEWS18 García-Durán et al. (2018), ICEWS05-15 Jin et al. (2020a) and YAGO Mahdisoltani et al. (2015). However, traditional benchmark datasets lack a comprehensive evaluation of accuracy, generalization, and robustness, which according to transductive, inductive, and few-shot scenarios. Therefore, we design three datasets based on the ACLED (The Armed Conflict Location & Event Data Project)[2] and ICEWS: ACLED2023, ACLED-IND, and ICEWS14-FS. These datasets aim to address specific issues in existing transductive, inductive, and few-shot datasets:

- **ACLED2023**: To address the potential issue of information leakage due to the outdated nature of existing datasets, where PLMs may learn from historical event texts, we compiled all events from 2023 available in the ACLED database to create ACLED2023. This mitigates the risk of information leakage since the release date of PLMs like ALBERT was in 2020, well before the events in ACLED2023 occurred.

- **ACLED-IND**: Traditional approaches for partitioning inductive datasets typically segment the entity set such that $(\mathcal{E}_{train} \cap \mathcal{E}_{valid} \cap \mathcal{E}_{test} = \emptyset)$, while ensuring that $(\mathcal{R}_{valid} \in \mathcal{R}_{train})$ and $(\mathcal{R}_{test} \in \mathcal{R}_{train})$. However, this can disrupt the temporal and topological features of TKGs. For instance, in the ICEWS14 dataset, *China* and *United States* are among the most frequently occurring entities. If they are categorized in the same set, it could result in many related facts being excluded from other sets, severely affecting the graph's connectivity and the capture of temporal features. Therefore, we retrieved events from ACLED occurring in Asia between 2019-2022 for the training set, and events from 2023 in Europe and the Americas for the validation and test sets, respectively. This design ensures that $(\mathcal{E}_{train} \cap \mathcal{E}_{valid} = \emptyset)$ and $(\mathcal{E}_{train} \cap \mathcal{E}test = \emptyset)$, maintaining the structural integrity of the training $[(\mathcal{G}_{train})]$, validation $[(\mathcal{G}_{valid})]$, and test $[(\mathcal{G}_{test})]$ graphs.

- **ICEWS14-FS**: To address the lack of exploration in few-shot scenarios in existing benchmark datasets, we created the ICEWS14-FS dataset based on the ICEWS14 dataset by extracting 10% of events at each timestamp. This process is designed to test the performance of models in information-sparse environments.

For a comprehensive evaluation of our CoLR, we utilized classic link prediction metrics, including MRR and Hits@k, to assess our framework's performance across various tasks. Details on the definitions and calculation methods of these metrics can be found in Appendix C.1. Additionally, a comparison of the datasets is provided in Appendix C.3.

---

[2] https://acleddata.com/

Table 1: Comparison results for transductive and few-shot settings on ICEWS14, ICEWS18, ICEWS05-15, ACLED2023 and ICEWS14-FS.

| Dataset | Metrics | TTransE | TA-DisMult | RE-GCN | TiRGN | CENET | HGLS | RPC | TLogic | ALRE-IR | ILR-IR | CoLR | + Improve(%) |
|---|---|---|---|---|---|---|---|---|---|---|---|---|---|
| ICEWS14 | MRR | 12.86 | 26.22 | 37.78 | 44.04 | 41.30 | 40.06 | 44.55* | 40.42 | 54.01† | 52.42 | **75.72** | 21.71 |
| | Hits@1 | 3.14 | 16.83 | 27.17 | 33.83 | 32.58 | 28.69 | 34.87* | 32.11 | 42.79† | 38.23 | **66.12** | 23.33 |
| | Hits@3 | 15.72 | 29.72 | 42.50 | 48.95 | 46.36 | 45.70 | 49.80* | 45.41 | 61.16† | 60.42 | **83.31** | 22.15 |
| | Hits@10 | 33.65 | 45.23 | 58.84 | 63.84 | 58.22 | 64.80 | 65.08* | 56.09 | 71.79 | 80.43† | **91.49** | 11.06 |
| ICEWS18 | MRR | 8.44 | 16.42 | 27.51 | 33.66 | 29.65 | 26.66 | 34.91* | 28.41 | 38.41† | 35.94 | **68.74** | 30.33 |
| | Hits@1 | 1.85 | 8.60 | 17.82 | 23.19 | 19.98 | 16.14 | 24.34* | 18.74 | 25.66† | 22.16 | **61.29** | 35.63 |
| | Hits@3 | 8.95 | 18.13 | 31.17 | 37.99 | 33.72 | 29.74 | 38.74* | 32.71 | 43.72† | 42.05 | **76.57** | 32.84 |
| | Hits@10 | 22.38 | 32.51 | 46.55 | 54.22 | 48.23 | 48.25 | 55.89* | 47.97 | 61.00 | 64.26† | **83.37** | 19.11 |
| ICEWS05-15 | MRR | 16.53 | 27.51 | 38.27 | 50.04 | 47.13 | 43.26 | 51.14* | 45.99 | 60.18† | 58.64 | **76.82** | 16.64 |
| | Hits@1 | 5.51 | 17.57 | 27.43 | 39.25 | 37.25 | 32.31 | 39.47* | 34.49 | 48.97† | 46.72 | **70.57** | 21.60 |
| | Hits@3 | 20.77 | 31.46 | 43.06 | 56.13 | 54.16 | 48.68 | 57.11* | 52.89 | 67.77† | 66.18 | **82.69** | 14.92 |
| | Hits@10 | 39.26 | 47.32 | 59.93 | 70.71 | 67.61 | 64.56 | 71.75* | 67.39 | 77.50 | 80.39† | **90.47** | 10.08 |
| YAGO | MRR | 32.57 | 54.92 | 82.30 | 87.95 | 61.83 | - | 88.87* | 83.61 | - | - | **94.23** | 5.36 |
| | Hits@1 | 27.94 | 48.15 | 78.83 | 84.34 | 48.02 | - | 85.10* | 83.82 | - | - | **93.40** | 8.30 |
| | Hits@3 | 43.39 | 59.61 | 84.27 | 91.37 | 57.92 | - | 92.57* | 84.51 | - | - | **94.48** | 1.91 |
| | Hits@10 | 53.37 | 66.71 | 88.58 | 92.92 | 80.06 | - | 94.04* | 83.85 | - | - | **95.43** | 1.39 |
| ACLED2023 | MRR | 36.59 | 66.41 | 73.15 | 73.82 | 71.16 | 76.97* | - | 75.55† | - | - | **83.24** | 6.27 |
| | Hits@1 | 19.73 | 51.09 | 63.39 | 63.59 | 62.11 | 66.48* | - | 67.65† | - | - | **74.76** | 7.11 |
| | Hits@3 | 47.96 | 80.67 | 80.72 | 82.00 | 77.89 | 85.58* | - | 81.79† | - | - | **90.24** | 4.66 |
| | Hits@10 | 67.20 | 88.81 | 90.38 | 91.32* | 91.18 | 87.00 | - | 89.13† | - | - | **97.62** | 6.30 |
| ICEWS14-FS | MRR | 2.37 | 21.37 | 11.08 | 32.87* | 30.57 | 30.87 | - | 22.92† | - | - | **50.95** | 18.08 |
| | Hits@1 | 0.08 | 16.68 | 6.99 | 23.87 | 24.44* | 18.99 | - | 17.81† | - | - | **40.68** | 16.24 |
| | Hits@3 | 0.21 | 23.37 | 12.52 | 36.68* | 33.23 | 34.04 | - | 25.95† | - | - | **54.92** | 18.24 |
| | Hits@10 | 6.39 | 30.07 | 17.85 | 49.55 | 43.31 | 56.51* | - | 32.29† | - | - | **72.78** | 16.27 |

Note: The best results are highlighted in **bold**. The symbols ⋆ and † denote the best results within representation-based and rule-based methods, respectively.

**Baselines.** We compare CoLR with representation-based methods:TTransE Leblay & Chekol (2018), TA-DisMult García-Durán et al. (2018), RE-GCN Li et al. (2021), TiRGN Li et al. (2022a), CENETXu et al. (2023), HGLS Zhang et al. (2023a) and RPCLiang et al. (2023), as well as rule-based methods: TLogic Liu et al. (2022), ALRE-IR Mei et al. (2022), and ILR-LR Mei et al. (2024). Except for CENET, the experimental results of all baselines on the existing benchmark datasets were taken from prior papers. Considering that CENET's experimental setup differs from other baselines, we reproduced its results on the ICEWS and YAGO datasets under the time-filtering setting Li et al. (2022a); Liang et al. (2023) for a fair comparison. For the proposed datasets, we conducted experiments for each baseline using their parameter settings on ICEWS14 and reported the results. The implementation details of our CoLR are provided in Appendix C.2.

## 6.2 MAIN RESULTS

The experimental results presented in Tables 1 and Tables 2 provide a detailed comparative analysis of various machine learning models across transductive, few-shot, and inductive scenarios. For comprehensive comparison and analysis of inductive capabilities, please refer to Appendix C.4.

**Transductive and Few-shot Analysis (Table 1)** As shown in Table 1, our CoLR significantly outperforms existing baseline models across all metrics on four transductive datasets, demonstrating its robust ability to integrate structured and temporal data effectively. For example, for ICEWS14, CoLR achieves 21.71% improvement over the next best result. Compared to the sub-optimal rule-based model ALRE-IR, CoLR shows improvements of 21.17%, 30.33%, and 16.64% in MRR across the three ICEWS datasets, respectively. These performance gains are attributed to the high-quality paths extracted and the designed multi-hop path encoder. Our method also surpasses both representation-based methods and discrete logical reasoning approaches in comprehensive performance, demonstrating that the introduction of path embeddings can significantly enhance reasoning capabilities. Particularly on the ICEWS18 dataset, where the density of events at single timestamps makes it challenging to capture structural features, temporal characteristics, and logical rules, other methods' performance drops sharply. Our model maintains stable performance in this complex scenario by integrating structural information, logical correlations, textual semantics, and temporal features. Additionally, our model performs exceptionally well on the ACLED2023 dataset, indicating effective use of the encoding capabilities of PLM rather than solely relying on its prior knowledge.

In the few-shot scenario, due to the lack of usable connected paths, traditional multi-hop logical reasoning methods like TLogic see a significant decline in reasoning performance on the ICEWS14-FS dataset. However, our CoLR excels with its MRR exceeding the next best model, TiRGN, by

Table 2: Inductive results of CoLR on four datasets.

| Test | ICEWS14 | | | ICEWS18 | | | ICEWS05-15 | | | ACLED-IND |
|---|---|---|---|---|---|---|---|---|---|---|
| Train | ICEWS14 | ICEWS18 | ICEWS05-15 | ICEWS18 | ICEWS14 | ICEWS05-15 | ICEWS05-15 | ICEWS14 | ICEWS18 | |
| MRR | 75.72 | 74.42 | 72.96 | 68.74 | 61.59 | 61.24 | 76.82 | 78.69 | 77.96 | 83.63 |
| Hits@1 | 66.12 | 64.37 | 64.37 | 61.29 | 54.08 | 53.58 | 70.57 | 73.42 | 71.52 | 80.71 |
| Hits@3 | 83.31 | 81.88 | 78.90 | 76.57 | 68.07 | 67.57 | 82.69 | 82.80 | 83.76 | 84.88 |
| Hits@10 | 91.49 | 90.09 | 86.52 | 83.37 | 77.30 | 76.58 | 90.47 | 91.30 | 91.78 | 88.81 |

18.08%. This performance indicates that CoLR captures relevant historical paths and learns the historical structural information of nodes to adapt to few-shot scenarios, thereby achieving remarkable performance.

**Inductive Analysis. (Table 2)** We train CoLR on one dataset and then test it on multiple others to demonstrate its cross-dataset application capabilities. The inductive results further highlight CoLR's robustness, as it consistently performs well across different training and testing dataset combinations. For instance, when trained ICEWS14 and tested on ICEWS05-15, it achieved an MRR of 78.69%; conversely, when trained on ICEWS05-15 and tested on ICEWS14, its MRR was 72.96%, showing only a slight performance variation. These outcomes illustrate CoLR's ability to generalize beyond its immediate training conditions, demonstrating that our model can deeply capture and effectively utilize the logical rules within TKGs, maintaining high prediction quality across various datasets.

## 6.3 ABLATION STUDY

The experimental results presented in Table 3 showcase the effectiveness of various components within the CoLR model, illustrated through an ablation study conducted on the ICEWS14 dataset. We undertake a more comprehensive analysis of the proposed method in Appendix C.5 and Appendix C.6, encompassing scalability, parameter sensitivity, and explainability.

**Temporal Relation Structure Graph Ablation (CoLR$_{-TRSG}$).** Removing the TRSG component resulted in a decrease across all metrics (MRR, Hits@1, Hits@3, Hits@10) by 2.97% compared to the full CoLR model. This substantial drop highlights the TRSG's crucial role in leveraging logical correlations and temporal cohesion to guide efficient pathfinding in TKGs.

**Time Sequence Encoder Ablation (CoLR$_{-TSE}$).** After replacing the TSE with the time-point encoder used in ALP-IR, performance significantly declined. This result confirms that traditional time-point encoders only model the temporal relations between historical paths and query tuples, without capturing temporal features. In contrast, our Time Sequence Encoder (TSE) can combine time sequences with logical rules in a more detailed manner, significantly enhancing the model's prediction accuracy.

Table 3: Abalation Studies of CoLR under ICEWS14.

| Model | ICEWS14 | | | |
|---|---|---|---|---|
| | MRR | Hits@1 | Hits@3 | Hits@10 |
| CoLR$_{-TRSG}$ | 72.93 | 62.34 | 81.28 | 87.95 |
| CoLR$_{-TSE}$ | 68.95 | 61.43 | 73.09 | 80.71 |
| CoLR$_{-PSS}$ | 63.23 | 57.74 | 64.17 | 73.09 |
| CoLR$_{-RP}$ | 71.29 | 64.63 | 72.36 | 79.26 |
| CoLR | 75.72 | 66.12 | 83.31 | 91.49 |

**Path Supplement Strategy Ablation (CoLR$_{-PSS}$).** During the path searching process, for missing paths, we solely use the textual sequences of the subject and object as representations of the historical paths, substituting the Path Supplement Strategy. The results for variant-PSS in the ablation study significantly decreased, demonstrating the positive effect of the Path Supplement Strategy in enhancing model performance.

**Multi-hop Path Ablation (CoLR$_{-RP}$).** After replacing the full multi-hop paths containing entities with paths containing only relations, the model's performance on all metrics decreased by an average of 7.28%. This indicates that full path encoding can capture critical semantic knowledge and structural information that cannot be obtained through relation paths alone, thus enhancing the model's ability to handle complex temporal reasoning.

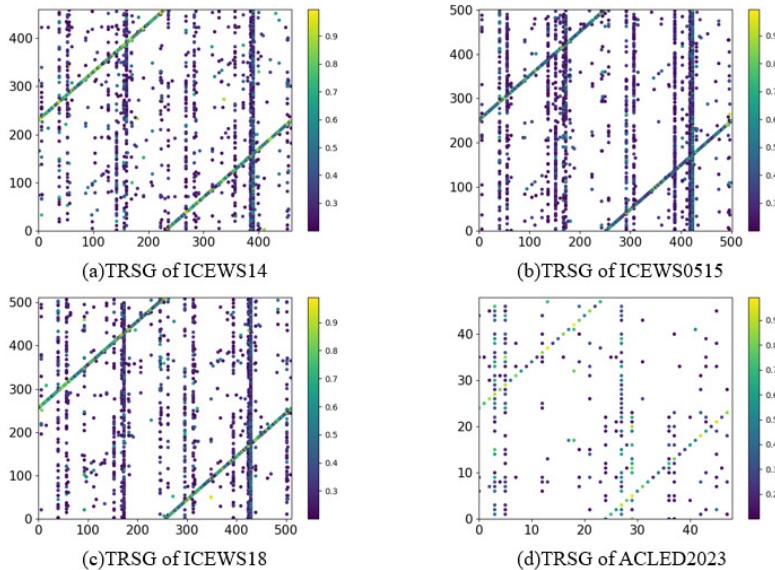

Figure 4: Visualization of TRSG from ICEWS14, ICEWS18, ICEWS05-15 and ACLED2023 datasets.

### 6.4 VISUALIZATION OF TRSG

We visualize the TRSG constructed from ICEWS14, ICEWS18, ICEWS05-15, and ACLED2023 to provide an intuitive analyze of their impact on temporal logical reasoning. As shown in Figure 4, we present the edges with strong cohesion in the TRSG ($>0.2$ for ICEWS datasets and $>0.1$ for ACLED2023) using scatter plots. It can be observed that there are two distinct diagonal lines in the four sub-figures of Figure 4, indicating that relations and their inverses typically exhibit strong cohesion in TKGs. In other words, there is a substantial amount of repetitive events in TKGs, supporting the basic assumption of previous works Li et al. (2022a); Mirtaheri et al. (2023); Liang et al. (2023). Additionally, in the ICEWS datasets, there are certain relations that evidently exhibit cohesion with the majority of other relations. These relations often appear with high frequency in the dataset and serve as conjunctions in temporal logical paths. Notably, despite the different numbers of relations in the three ICEWS datasets, their cohesion distributions are strikingly similar. This substantiates that the TRSG we propose can effectively capture stable structure information of relations in the datasets and enhance the robustness of logical reasoning in inductive scenarios.

## 7 CONCLUSIONS AND FUTURE WORK

In this work, we propose a two-stage coherent logical reasoning framework, named CoLR, which integrates structural dependencies and text semantics of temporal paths to tackle the link forecasting task over TKGs. It searches for temporal logical paths in the first stage and encodes these paths using a PLM and a novel time sequence encoder in the second stage. To efficiently search for reliable paths, we construct TRSG based on cohesion between relations and design a temporal fusion search graph. To address path-missing issues, we implement a path supplement strategy that samples historical paths based on cohesion. Additionally, we introduced three new datasets to address the deficiencies of existing benchmarks in transductive, inductive, and few-shot scenarios. Experimental results demonstrate that CoLR comprehensively outperforms existing methods across six datasets. Since temporal paths consist of only 1 to 3 hop edges, the corresponding text sequences pose challenges for PLMs in capturing semantic dependencies. In future work, we plan to integrate LLMs to generate more detailed and precise texts to enhance the ability of PLMs to deeply understand the latent semantics of text sequences.

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

# A METHODS

## A.1 TIME-FUSION ALGORITHM

The time-fusion algorithm is proposed to accurately search for reliable logical paths. The process is shown in Algorithm 1. Intuitively, Algorithm 1 is a breadth-first and depth-first algorithm, where $\mathcal{N}_{tf}[u]$ denotes the neighbors of $u$ in $\mathcal{G}_{tf}$ with previous timestamps. $generatePath$ reverse the edge list $\{([o, t_q], r_l, [e_{l-1}, t_l]), ..., ([e_1, t_1], r_1, [s, t_1])\}$ to generate a temporal path $\{(s, r_1, e_1, t_1), ..., (e_{l-1}, r_l, t_1)\}$. $sampleEdge$ samples the next edge based on both time intervals and relation cohesions, *i.e.* $P_{next} = P_{time} + P_{coh}$. $PSS$ denotes path supplement strategy.

---

**Algorithm 1** Time-fusion Algorithm

---

**Input:** time-fusion search graph $\mathcal{G}_{tf}$, TRSG with cohesion $\hat{\mathbf{R}}_{coh}$, query quadruplet $(s, r_q, o, t_q)$, max path length $L$, max path number $K$
**Output:** list of extracted logical paths $\mathbb{P}$
 1: Init $\mathbb{P} = List()$; $q = Stack()$; $visited = Set([o, t_q])$; $prev = Dict()$;
 2: **if** $|\mathcal{G}_{tf}|$=0 **then**
 3:     $\mathbb{P}$.append($PSS(\mathcal{G}_{tf}, s, o)$)
 4: **end if**
 5: $q$.push($[o, t_q]$); $visited[[o, t_q]] = True$
 6: **while** $|q| > 0$ **do**
 7:     $u = q[-1]$
 8:     **if** $u$ not in $\mathcal{G}_{tf}$ or $|q| >= L$ **then**
 9:         $q$.pop();continue
10:     **end if**
11:     **if** $u$ not in $prev$ **then**
12:         $prev[u] = \mathcal{N}_{tf}[u]$
13:     **end if**
14:     $nextedges$ = $prev[u]$
15:     **for** $edge$ in $nextedges$ **do**
16:         **if** $edge[2][0] == s$ **then**
17:             $\mathbb{P}$.append($generatePath(q, edge)$)
18:         **end if**
19:         **if** $|\mathbb{P}| >= L$ **then**
20:             break
21:         **end if**
22:     **end for**
23:     **if** $|\mathbb{P}| >= L$ **then**
24:         break
25:     **end if**
26:     $nextedge = sampleEdge(nextedges)$; $q$.append($nextedge$); $prev[u]$.delete($nextedge$)
27: **end while**
28: **if** $|\mathbb{P}|$=0 **then**
29:     $\mathbb{P}$.append($PSS(\mathcal{G}_{tf}, s, o)$)
30: **end if**

---

## A.2 APPLICATIONS OF TRSG

The components of the proposed CoLR framework can be flexibly transferred to other logical reasoning methods. For instance, for neural logical reasoning approaches, TRSG and TFSG can optimize the efficiency of temporal path search, enhancing the reliability of logical pathways. Additionally, the time sequence encoder can help capture more nuanced temporal information, thereby improving reasoning performance.

For symbolic methods, we propose a novel cohesion-based rule confidence evaluation function as follows.

**Definition 2.** *Given a query quadruplet $(s, r_q, o, t_q)$, the subject and object cohesion confidence of a relation path $\Upsilon_l = (r_1, r_2, ..., r_l, r_q)$ is defined as follows:*

$$conf_{coh}^s(\Upsilon_l) = \frac{1}{l+1} \sum_{i=0}^{i=l-1} (1 - exp(-\hat{\mathbf{R}}_{coh}^\omega[r_i, r_{i+1}])), \tag{10}$$

$$conf_{coh}^o(\Upsilon_l) = \frac{1}{l+1} \sum_{i=1}^{i=l} (1 - exp(-\hat{\mathbf{R}}_{coh}^\omega[r_i, r_{i+1}])). \tag{11}$$

The formulation 10 and 11 are designed to calculate the weight of a coherent path composed of $B_{\Upsilon_l} = (r_1, r_2, ..., r_l)$ and $H_{\Upsilon_l} = r_q$ in the relation path, where $\hat{\mathbf{R}}_{coh}^\omega[r_0, r_1] = \hat{\mathbf{R}}_{coh}^\omega[r_1^-, r_q]$ and $\hat{\mathbf{R}}_{coh}^\omega[r_l, r_{l+1}] = \hat{\mathbf{R}}_{coh}^\omega[r_l, r_q^-]$, respectively. A high-confidence path is expected to exhibit stronger cohesion. We then combine the two cohesion confidence scores to obtain the final confidence score:

$$conf_{coh}(\Upsilon_l) = \frac{1}{2}(conf_{coh}^s(\Upsilon_l) + conf_{coh}^o(\Upsilon_l)). \tag{12}$$

Compared to traditional confidence evaluation approaches, cohesion confidence is more intuitive and efficient. For instance, the evaluation function in TLogic can be formalized as

$$conf_{sup}(B_{\Upsilon_l} \Rightarrow H_{\Upsilon_l}) = \frac{sup(B_{\Upsilon_l} \Rightarrow H_{\Upsilon_l}(x, y, t))}{\#(x, y, B_{\Upsilon_l}) : \exists z_1, ..., z_l; t_1, ..., t_l : B_{R_l}}, \tag{13}$$

where $sup(\cdot)$ denotes the count of quadruplets with $H_{\Upsilon_l}$ supported by $B_{\Upsilon_l}$, $\#(x, y, B_{\Upsilon_l})$ is the number of unique paths grounded by $B_{\Upsilon_l}$. Clearly, confidence evaluation approaches similar to $conf_{sup}(B_{\Upsilon_l} \Rightarrow H_{\Upsilon_l})$ require extensive path sampling for each rule. In contrast, once the TRSG of a TKG is constructed, our cohesion-based method can directly estimate the confidence of a rule. Experimental results indicate that our confidence estimation function shows only a slight performance decrease compared to $conf_{sup}(B_{\Upsilon_l} \Rightarrow H_{\Upsilon_l})$ when applied to TLogic.

### A.3 COMPUTATIONAL COMPLEXITY OF TRSG

As shown in Equation 2, the TRSG on each subgraph can be obtained by multiplying two entity-relation matrices, which is efficiently implemented in PyTorch. Therefore, as the graph size expands, the computational cost mainly arises from the increased number of matrix multiplications due to the extension of the timestamp sequence. For a TKG with $N$ timestamps, when the time window $\omega$ is 1, the times of matrix multiplications is $N$; when the time window size is $N$, the computation times is $1/2(N*(N+1))$. Thus, the time complexity for constructing a TRSG lies between $O(N)$ and $O(N^2)$. Since the time window is typically much smaller than $N$, the time complexity of constructing TRSG approaches $O(N)$. Clearly, as the number of timestamps in the TKG increases, the computational cost does not grow dramatically. In fact, for the ICEWS05-15 dataset, which contains over 4000 timestamps, constructing a TRSG with a time window of 10 takes only 20 seconds. In summary, our proposed TRSG can flexibly scale to very large real-world graphs.

## B PROOFS

### B.1 PROOF OF THEOREM 1

*Proof.* Let $(e_1, r_1, e_2, t_i) \in \mathcal{G}_i$ and $(e_2, r_2, e_3, t_j) \in \mathcal{G}_j$ represent any two coherent edges in the TKG, where $\mathcal{G}_i$ and $\mathcal{G}_j$ are subgraphs and $|t_i - t_j| \le \omega$. $\mathbf{E}_o^i \in \mathbb{R}^{|\mathcal{E}| \times |\mathcal{R}|}$ is the object-relation matrix of $\mathcal{G}_i$ and $\mathbf{E}_s^j \in \mathbb{R}^{|\mathcal{E}| \times |\mathcal{R}|}$ is the subject-relation matrix of $\mathcal{G}_j$. To calculate the cohesion matrix, we first need to discuss the three cases of connectivity in the TKG:

(I) Consider the case where $t_i = t_j$. In this case, the two edges are in the same subgraph $\mathcal{G}_i$ ($\mathcal{G}_i = \mathcal{G}_j$). According to Section 4.1, the cohesion matrix in $\mathcal{G}_i$ can be calculated as $\mathbf{R}_{coh} = \mathbf{E}_o^{i^T} \mathbf{E}_s^i$. Then, the cohesion between $r_1$ and $r_2$ in $\mathcal{G}_i$ is $c_{12}^i = \mathbf{R}_{coh}^i[1, 2]$.

(II) Consider the case where $t_i < t_j$. The two edges are coherent from a historical graph $\mathcal{G}_i$ to a future graph $\mathcal{G}_j$ through $e_2$. Therefore, the cohesion matrix from $\mathcal{G}_i$ to $\mathcal{G}_j$ is calculated as $\mathbf{R}_{coh} = \mathbf{E}_o^{i^T} \mathbf{E}_s^j$. Then we have $c_{12}^{ij} = \mathbf{R}_{coh}[1, 2]$, where $c_{12}^{ij}$ is the cohesion between $r_1$ and $r_2$.

(III) Consider the case where $t_i > t_j$. Although these two edges are structurally coherent, they violate temporal constraints and cannot serve as part of a temporal logical path. Therefore, such interactions between relations should be ignored.

By summing all the (I) intra-subgraph and (II) inter-subgraph cohesion matrices in a TKG, we can obtain the overall TKG relation cohesion matrix. Then, the formulation in Theorem 1 can be derived through mathematical induction:

**Base Case ($\omega = 1$):** Let $\mathbf{R}_{coh}^{\omega}$ denote the cohesion matrix with a time window $\omega$. For $\omega = 1$, the cohesion matrix of $\mathcal{G}$ is obtained by summing the cohesion matrices within each subgraphs, thus $\mathbf{R}_{coh}^1 = \sum_{i=i}^{k} \mathbf{E}_o^{i^T} \mathbf{E}_s^i$.

**Inductive hypothesis:** Assume that for some positive integer $1 < k \le |\mathcal{T}| - 1$, the formula

$$\mathbf{R}_{coh}^k = \sum_{j=k}^{|\mathcal{T}|} (\sum_{i=j-k+1}^{j} \mathbf{E}_o^{i^T}) \mathbf{E}_s^j + \sum_{j=1}^{k-1} (\sum_{i=1}^{j} \mathbf{E}_o^{i^T}) \mathbf{E}_s^j$$

holds true when $\omega = k$.

**Inductive step ($\omega = k + 1$):** Consider that the extension of the time window will introduce new items $\sum_{i=1}^{|\mathcal{T}|-\omega+1} \mathbf{E}_o^{i^T} \mathbf{E}_s^{i+\omega-1}$, we add them to $\mathbf{R}_{coh}^k$:

$$\mathbf{R}_{coh}^{\omega} = \mathbf{R}_{coh}^k + \sum_{i=1}^{|\mathcal{T}|-\omega+1} \mathbf{E}_o^{i^T} \mathbf{E}_s^{i+\omega-1}$$

$$= \sum_{j=k}^{|\mathcal{T}|} (\sum_{i=j-k+1}^{j} \mathbf{E}_o^{i^T}) \mathbf{E}_s^j + \sum_{j=1}^{k-1} (\sum_{i=1}^{j} \mathbf{E}_o^{i^T}) \mathbf{E}_s^j + \sum_{i=1}^{|\mathcal{T}|-k} \mathbf{E}_o^{i^T} \mathbf{E}_s^{i+k}$$

$$= \sum_{j=k+1}^{|\mathcal{T}|} (\sum_{i=j-k+1}^{j} \mathbf{E}_o^{i^T}) \mathbf{E}_s^j + \sum_{j=k}^{k} (\sum_{i=1}^{j} \mathbf{E}_o^{i^T}) \mathbf{E}_s^j + \sum_{j=1}^{k-1} (\sum_{i=1}^{j} \mathbf{E}_o^{i^T}) \mathbf{E}_s^j + \sum_{j=k+1}^{|\mathcal{T}|} \mathbf{E}_o^{j-k^T} \mathbf{E}_s^j$$

$$= \sum_{j=k+1}^{|\mathcal{T}|} (\sum_{i=j-k}^{j} \mathbf{E}_o^{i^T}) \mathbf{E}_s^j + \sum_{j=1}^{k} (\sum_{i=1}^{j} \mathbf{E}_o^{i^T}) \mathbf{E}_s^j$$

$$= \sum_{j=\omega}^{|\mathcal{T}|} (\sum_{i=j-\omega+1}^{j} \mathbf{E}_o^{i^T}) \mathbf{E}_s^j + \sum_{j=1}^{\omega-1} (\sum_{i=1}^{j} \mathbf{E}_o^{i^T}) \mathbf{E}_s^j.$$

Consider the extreme case of $k = |\mathcal{T}| - 1$, the cohesion matrix can be calculated by summing the cohesion matrices within all subgraphs and the cohesion matrices between any two increment subgraphs.

$$\mathbf{R}_{coh}^{\omega} = \sum_{j=1}^{|\mathcal{T}|} \mathbf{E}_o^{j^T} \mathbf{E}_s^j + \sum_{j=2}^{|\mathcal{T}|} (\sum_{i=1}^{j-1} \mathbf{E}_o^{i^T}) \mathbf{E}_s^j$$

$$= \sum_{j=|\mathcal{T}|}^{|\mathcal{T}|} (\sum_{i=j}^{j} \mathbf{E}_o^{i^T}) \mathbf{E}_s^j + \sum_{j=1}^{|\mathcal{T}|-1} \sum_{i=j}^{j} \mathbf{E}_o^{i^T} \mathbf{E}_s^j + \sum_{j=|\mathcal{T}|}^{|\mathcal{T}|} (\sum_{i=1}^{j-1} \mathbf{E}_o^{i^T}) \mathbf{E}_s^j + \sum_{j=2}^{|\mathcal{T}|-1} (\sum_{i=1}^{j-1} \mathbf{E}_o^{i^T}) \mathbf{E}_s^j$$

$$= \sum_{j=|\mathcal{T}|}^{|\mathcal{T}|} (\sum_{i=1}^{j} \mathbf{E}_o^{i^T}) \mathbf{E}_s^j + \sum_{j=2}^{|\mathcal{T}|-1} (\sum_{i=1}^{j} \mathbf{E}_o^{i^T}) \mathbf{E}_s^j + \sum_{j=1}^{1} (\sum_{i=1}^{j} \mathbf{E}_o^{i^T}) \mathbf{E}_s^j$$

$$= \sum_{j=|\mathcal{T}|}^{|\mathcal{T}|} (\sum_{i=1}^{j} \mathbf{E}_o^{i^T}) \mathbf{E}_s^j + \sum_{j=1}^{|\mathcal{T}|-1} (\sum_{i=1}^{j} \mathbf{E}_o^{i^T}) \mathbf{E}_s^j$$

$$= \sum_{j=\omega}^{|\mathcal{T}|} (\sum_{i=j-\omega+1}^{j} \mathbf{E}_o^{i^T}) \mathbf{E}_s^j + \sum_{j=1}^{\omega-1} (\sum_{i=1}^{j} \mathbf{E}_o^{i^T}) \mathbf{E}_s^j.$$

Thus, for any positive integer time window $\omega \le |\mathcal{T}|$, Formula 2 holds true. Proof completed. $\qquad\square$

# C  EXPERIMENTS

## C.1  EVALUATION METRICS

Link forecasting is the main objective of our experiments. For each query quadruplet $(s, r_q, o, t)$ in $\mathcal{G}_{test}$, we generate two queries $(s, r_q, ?, t_q)$ and $(o, r_q^-, ?, t_q)$ for answering $o$ and $s$, respectively. Following the setup of previous work Zha et al. (2022); Su et al. (2024), for each query quadruplet, we sample 49 negative and 1 positive candidates entities from the past $\omega$ subgraphs in evaluation. Temporal paths are extracted by constructing quadruplets for each candidate entity, from which scores are calculated. The rank list of quadruplet scores is then utilized to assess the effectiveness of reasoning by the Mean Reciprocal Rank (MRR for short) and Hits@k metrics.

## C.2  IMPLEMENTATION DETAILS

For our experiments, we tailor the $\omega$ parameter to suit the characteristics of each dataset. Specifically, for ICEWS14, ICEWS18, ICEWS14-FS and YAGO datasets, $\omega$ is set to 5; for ICEWS05-15, ACLED2023, and ACLED-IND datasets, it's set to 10. Additionally, we set the maximum path length (L) and the number of paths (K) across all datasets to 3. The hyperparameters $\delta$ and $\gamma$ are fixed to 0.05 and 0.1, respectively. Following previous works Li et al. (2022a); Liu et al. (2022); Mei et al. (2022), we perform quadruplet filtering under the time-filtering setting and only predicted future events at the next timestamp.

We implement our CoLR in a PyTorch 1.9.1 environment. The model is initialized using the "albert-base-v2"[3] pre-trained language model from Hugging Face, which outputs embeddings with a dimension of 768, consistent with the dimensions of the GRU units. The entire model is trained using the Adam optimizer with a learning rate of 1e-5. All experiments are conducted on a single NVIDIA 4090 GPU with 24GB of RAM.

## C.3  STATISTIC OF DATASETS

Table 4: Statistics information of six datasets, where # represent the number of the item.

| Dataset | ICEWS14 | ICEWS18 | ICEWS05-15 | ICEWS14-FS | ACLED2023 | ACLED-IND |
|---|---|---|---|---|---|---|
| #Entities | 7,128 | 23,033 | 10,488 | 2,955 | 4,850 | 3,464 |
| #Relations | 230 | 256 | 251 | 164 | 24 | 23 |
| #Train Facts | 63,685 | 373,108 | 322,958 | 6,368 | 57,076 | 45,787 |
| #Valid Facts | 13,823 | 45,995 | 69,224 | 1,382 | 12,031 | 6,781 |
| #Test Facts | 13,222 | 49,545 | 69,147 | 1,322 | 11,150 | 13,683 |
| #Timestamps | 365 | 304 | 4,014 | 365 | 365 | 1,826 |

## C.4  INDUCTIVE COMPARISON RESULTS

We compared CoLR with multi-hop logical reasoning methods in an inductive setting, and the experimental results are presented in Table 5. It can be observed that even when applied to datasets from different time periods, the performance of the three models only slightly decreased, demonstrating that logical rules are indeed crucial for solving inductive reasoning problems. When trained on ICEWS14 and ICEWS05-15 and tested on ICEWS18, the performance of the three methods declined most significantly because the fact distribution in ICEWS14 and ICEWS05-15 is sparser than in ICEWS18. Notably, despite some performance decline, our model still significantly outperforms ILR-IR in inductive scenarios, underscoring the robustness and superiority of our proposed framework. Furthermore, on the ACLED-IND dataset, our model exceeds TLogic by 22.24% in the MRR metric, showcasing its superior inductive capability and reasoning ability compared to TLogic.

---

[3]https://huggingface.co/albert/albert-base-v2/tree/main

Table 5: Comparison results for inductive setting on ICEWS14, ICEWS18, ICEWS05-15 datasets.

| Test | | | ICEWS14 | | | ICEWS18 | | | ICEWS05-15 | | | |
|------|------|------|---------|-----------|--------|---------|------------|------------|---------|---------|-----------|
| Train | | ICEWS14 | ICEWS18 | ICEWS05-15 | ICEWS18 | ICEWS14 | ICEWS05-15 | ICEWS05-15 | ICEWS14 | ICEWS18 | ACLED-IND |
| ILR-IR | MRR | 52.42 | 47.77 | 49.38 | 35.94 | 32.88 | 30.87 | 58.64 | 56.25 | 56.74 | - |
| | Hits@1 | 38.23 | 33.87 | 35.64 | 22.16 | 19.98 | 18.29 | 46.72 | 44.30 | 44.76 | - |
| | Hits@3 | 60.42 | 52.43 | 53.96 | 42.05 | 39.72 | 36.69 | 66.18 | 63.43 | 64.02 | - |
| | Hits@10 | 80.43 | 75.66 | 77.98 | 64.26 | 61.83 | 59.49 | 80.39 | 78.54 | 79.09 | - |
| CoLR | MRR | 75.72 | 74.42 | 72.96 | 68.74 | 61.59 | 61.24 | 76.82 | 78.69 | 77.96 | 83.63 |
| | Hits@1 | 66.12 | 64.37 | 64.37 | 61.29 | 54.08 | 53.58 | 70.57 | 73.42 | 71.52 | 80.71 |
| | Hits@3 | 83.31 | 81.88 | 78.90 | 76.57 | 68.07 | 67.57 | 82.69 | 82.80 | 83.76 | 84.88 |
| | Hits@10 | 91.49 | 90.09 | 86.52 | 83.37 | 77.30 | 76.58 | 90.47 | 91.30 | 91.78 | 88.81 |
| TLogic | MRR | 40.42 | 37.11 | 38.09 | 28.41 | 26.68 | 26.88 | 45.99 | 41.76 | 40.45 | 61.39 |
| | Hits@1 | 32.11 | 28.77 | 29.80 | 18.74 | 18.54 | 18.52 | 34.49 | 33.39 | 32.15 | 52.41 |
| | Hits@3 | 45.41 | 42.17 | 43.05 | 32.71 | 30.45 | 30.72 | 52.89 | 47.20 | 45.72 | 69.28 |
| | Hits@10 | 56.09 | 55.76 | 53.66 | 47.97 | 42.79 | 43.55 | 67.39 | 57.12 | 55.94 | 75.29 |

## C.5 COMPREHENSIVE ANALYSIS

In this section, we undertake a comprehensive analysis of the proposed method, encompassing scalability, parameter sensitivity, and explainability.

**Scalability Analysis.** To validate the scalability of our proposed two-stage architecture, we designed three variants: CoLR$_{+FRE}$, CoLR-GRU, and Tlogic$_{+TRSG}$. Here, $+FRE$ represents the integration of a frequency-based confidence evaluation method during historical path search; CoLR-GRU represents replacing the PLM in the CoLR framework with a traditional GRU sequence encoder; Tlogic$_{+TRSG}$ involves incorporating a temporal relation graph into TLogic to enhance path sampling efficiency. The specific implementation process is detailed in Appendix B, and scalability experiment results are shown in Table 6. CoLR$_{+FRE}$, by incorporating frequency information, showed performance improvement over the original model, demonstrating our model's ability to effectively combine other beneficial information through a logic-gated network to enhance reasoning accuracy. Meanwhile, TLogic$_{+TRSG(50)}$ performed significantly better than TLogic(50) and reached a level comparable to TLogic(200), where $(\cdot)$ denotes the sampling count; TLogic$_{+TRSG(200)}$ outperformed TLogic(200), indicating that TRSG can efficiently search logical paths using the cohesion among relations, reducing the computational cost of rule learning in TLogic while enhancing performance. Additionally, we observed that the performance of CoLR-GRU significantly surpassed ILR-IR (as shown in Table 2), thanks to high-quality path extraction ensuring even simple encoders could learn logical associations. However, there was a noticeable gap between CoLR-GRU and CoLR because GRU lacks the capability to recognize rich semantic information in text and faces insufficient modeling capacity when dealing with large-scale sequence data. These observations and results validate that our proposed methods can be easily merged or extended into existing diverse logical reasoning models.

Table 6: Scalability Analysis of CoLR under ICEWS14.

| Model | ICEWS14 | | | |
|-------|---------|--------|--------|---------|
| | MRR | Hits@1 | Hits@3 | Hits@10 |
| CoLR$_{+FRE}$ | 77.50 | 65.32 | 84.45 | 92.59 |
| CoLR-GRU | 67.98 | 61.10 | 70.35 | 81.37 |
| TLogic(50) | 38.52 | 30.47 | 43.69 | 53.51 |
| TLogic(200) | 40.42 | 32.11 | 45.41 | 56.09 |
| TLogic$_{+TRSG(50)}$ | 39.12 | 30.68 | 43.87 | 55.36 |
| TLogic$_{+TRSG(200)}$ | 41.17 | 32.99 | 45.79 | 57.03 |
| CoLR | 75.72 | 66.12 | 83.31 | 91.49 |

**Sensitivity Analysis.** During the search for historical paths, we conducted a comprehensive sensitivity analysis of our proposed method on ICEWS14, focusing particularly on the influence of the hyperparameters—time window $\omega$ and path count $K$—on model performance. To deeply understand

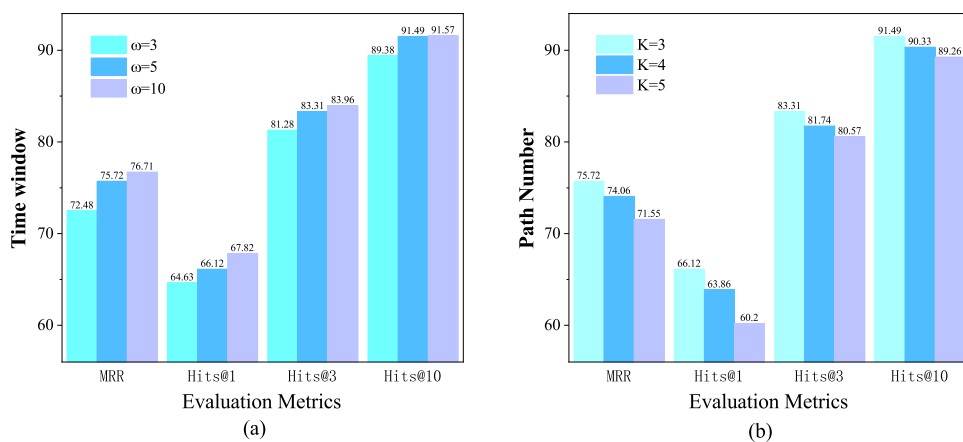

Figure 5: Sensitivity on (a) the time window $\omega$ and (b) the path number $K$ for CoLR.

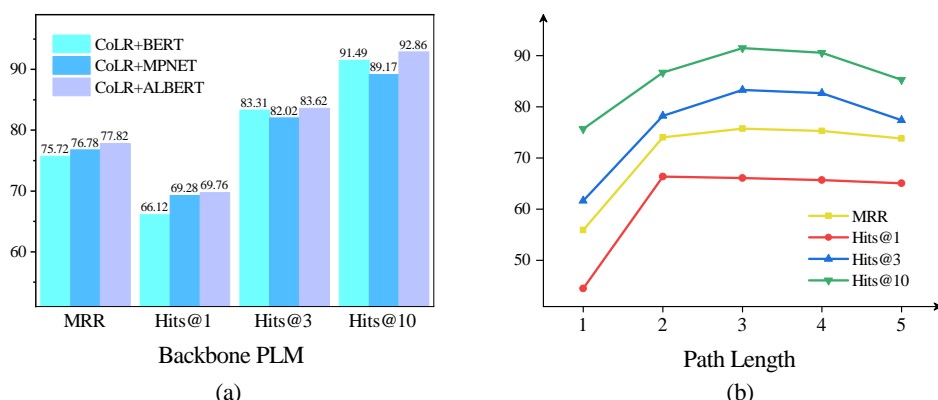

Figure 6: Sensitivity on (a) the backbone PLM and (b) the path length $L$ for CoLR.

the model's response to parameter changes, we adjusted these two critical parameters separately and experimentally verified their impact on reasoning performance. Figure 5(a) displays the specific effects of changes in the time window $\omega$ on performance. The results indicate that as the time window increases, the overall performance of the model also gradually improves. This is because a larger time window allows the model to consider events over a longer period, enhancing its ability to capture temporal relations, consistent with previous research findings Mei et al. (2022).

Additionally, we studied the impact of the variation in path count $K$ on performance, as shown in Figure 5(b). We found that the model achieves optimal performance when the number of paths is moderate. However, as the number of paths further increases, the inclusion of logically irrelevant paths also grows, introducing noise that leads to a gradual decline in model reasoning accuracy. This finding emphasizes the importance of quality over quantity in path selection and also explains why our method significantly surpasses previous methods in performance. These experiments not only showcase the direct impact of adjusting hyperparameters on model performance but also validate the robustness and adaptability of our approach under different parameter settings.

In Figure 6(b), we show the experimental results of our CoLR on ICEWS14 under different maximum path length constraints. When the maximum path length $L$ is set to 2, CoLR achieves the best performance on the Hits@1 metric. However, as the path length increases, the performance decreases. This suggests that while increasing the path search depth allows more paths to be retrieved, these paths are often logically unrelated to the query and may interfere with reasoning about the correct target entity. When the maximum path length is set to 3, CoLR achieves the best performance on the remaining three metrics. It indicates that a search range that is too small may prevent the correct

Table 7: Comparison of training time on ICEWS14 and ICEWS05-15.

| Dataset | TiRGN | HGLS | CoLR | CoLR-GRU |
|---|---|---|---|---|
| ICEWS14 | 25min | 17min | 2.3h | 22min |
| ICEWS05-15 | 4.5h | 3h | 12.8h | 3.5h |

Table 8: Comparison of path search time (in second) on ICEWS14, ICEWS18, and ICEWS05-15.

| Method | ICEWS14 | ICEWS18 | ICEWS05-15 |
|---|---|---|---|
| TLogic | 135 | 562 | 871 |
| CoLR | 38 | 156 | 192 |

entity from being retrieved. Taking these findings into account, we set the maximum path length $L$ to 3 for all other experiments.

To investigate the sensitivity of our CoLR to different PLM encoders, we constructed three variants using different PLMs as backbones: CoLR+ALBERT, CoLR+BERT, and CoLR+MPNET. We conducted experiments on ICEWS14 using these three variants, and the results are shown in Figure 6(a). It can be observed that the differences in MRR scores among the three variants are negligible. Although BERT Devlin et al. (2019) has larger model parameters compared to ALBERT Lan et al. (2020), CoLR+BERT only improves the MRR score by 1.06% compared to CoLR+ALBERT. When using MPNet Song et al. (2020), which has stronger sentence understanding capabilities, CoLR+MPNET achieves only a 1.04% improvement on MRR. These observations suggest that CoLR is a stable framework, and differences among PLMs can be effectively mitigated through subsequent fine-tuning.

**Efficiency Analysis.** Due to the utilization of the pre-trained language model, the training of CoLR is a time-intensive process. Despite our adoption of LoRA to enhance training efficiency, the training still consumes several hours. However, CoLR-GRU, employing GRU as the encoder, completes training in half an hour, rendering CoLR suitable for real-world applications.

We compared the training time of CoLR with other methods on ICEWS14 and ICEWS05-15 to further validate its training efficiency. For all baseline models, including our CoLR and CoLR-GRU, each model was trained until achieving its best performance. For a fair comparison, all models were trained on a single NVIDIA 4090 GPU. The experimental results are shown in Table 7. As observed, when using PLM as the encoder, our CoLR indeed requires a considerable amount of time for training. However, when replacing PLM with GRU, CoLR-GRU achieves comparable training speeds with other methods, even outperforming TiRGN. It demonstrates that the significant time cost of CoLR originates from fine-tuning the PLM. Notably, as shown in Table 6, CoLR-GRU still maintains excellent reasoning performance, significantly outperforming previous methods.

Additionally, we evaluated the path search time of CoLR on ICEWS14, ICEWS18, and ICEWS05-15 to investigate the efficiency of our path search approach. As shown in Table 8, the time cost of CoLR for path search is significantly lower than that of TLogic, validating the effectiveness of the proposed time-fusion path search algorithm. By introducing TRSG and TFSG, our time-fusion path search algorithm can efficiently identify historical paths relevant to the query. Specifically, TRSG ensures that CoLR only needs to retrieve the top-K historical paths, avoiding the additional computational cost of retrieving unrelated paths. TFSG preserves temporal information while compressing the TKG into a static graph, eliminating the need for re-expanding the static graph back into a TKG.

## C.6 CASE STUDY

**Explainability Analysis.** Compared to black-box representation-based methods, logical reasoning methods can provide reasonable explanations for predictions with logical rules. Our CoLR samples temporal paths from historical subgraphs, which naturally serve as the basis for reasoning. To analyze the explainability of our CoLR, we illustrate an inference example in Table 9 and Figure 7.

Table 9: Candidates and corresponding paths from fig 7 for query *(South Korea; Engage in diplomatic cooperation; ?; 2015-6-27)*

| Candidate | No. | Temporal path | Condifence |
|---|---|---|---|
| Xi Jinping | (1) | South Korea; Express intent to meet or negotiate⁻; Xi Jinping; 2015-6-23 | 0.73 |
| China | (2) | South Korea; Consult⁻; China; 2015-6-25 | 0.88 |
| | (3) | South Korea; Make an appeal or request; China; 2015-6-26 | 0.93 |
| Japan | (4) | South Korea; Criticize or denounce; Japan; 2015-6-23 | 0.16 |
| | (5) | South Korea; Reduce relations; Japan; 2015-6-22 | 0.11 |
| | (6) | South Korea; Reject; Japan; 2015-6-23 | 0.05 |

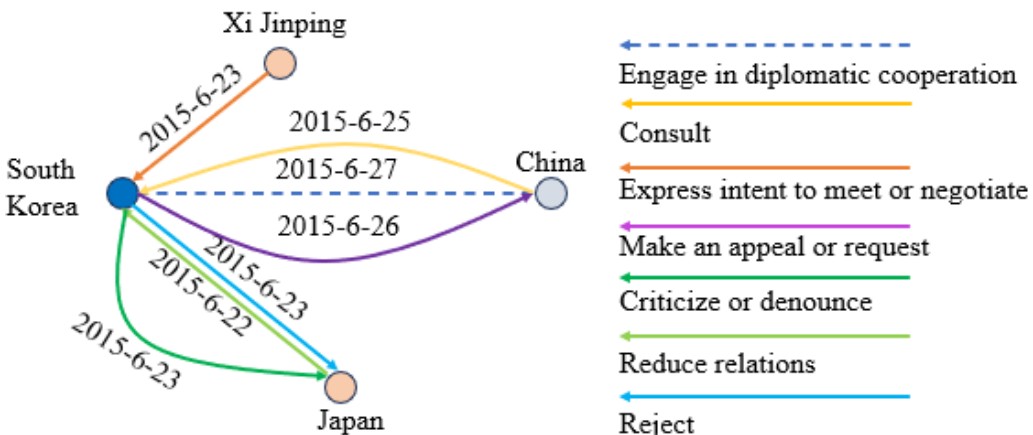

Figure 7: Historical interactions of *South Korea* for query *(South Korea; Engage in diplomatic cooperation; ?; 2015-6-27)* from ICEWS05-15 dataset.

Given a query quadruplet *(South Korea; Engage in diplomatic cooperation; ?; 2015-6-27)*, 6 paths in Table 9 can be extracted from its historical interactions in Figure 7. Obviously, the candidate *Japan* has the most frequent interactions with *South Korea* are the most frequent. However, the historical paths extracted from these interactions have little logical correlation with a relation *Engage in diplomatic cooperation*. Therefore, our CoLR assigns lower scores to paths (4), (5), and (6). For the intended object *China*, despite fewer interactions with *South Korea*, these interactions exhibit a remarkably close logical correlation with the query concerning both temporal intervals and semantics. As a result, path (2) and (3) achieved the highest confidence score. These findings and analyses substantiate the explainability and efficacy of our method.

Moreover, we observe that although *Xi Jinping* does not directly connect *South Korea* and *China* to form a valid path, the prior knowledge that *Xi Jinping* is the President of *China* allows path (1) to effectively assist in identifying the correct entity. This prior knowledge is derived from the advanced natural language understanding capabilities of the pre-trained language model and the structural information learned from the complete paths. This observation indirectly confirms the validity of our path supplement strategy.

