# OpenReview forum: "Integrating Relation Dependences and Textual Semantics for Coherent Logical Reasoning over Temporal Knowledge Graph"
_ICLR.cc/2025/Conference — Submitted to ICLR 2025_

### Official Review · Reviewer_pLWV · 2024-10-31

**Soundness:** 2
**Presentation:** 3
**Contribution:** 2
**Rating:** 5
**Confidence:** 3

**Summary:**

This paper proposes a time-fusion search method based on a temporal relationship structure graph to address the problem of existing TKGR models being difficult to effectively extract logical paths from temporal KGs. Afterwards, the joint pre-trained language model and GRU model is proposed to obtain more complete logical semantics in the logical path from the perspectives of logical context and time series, effectively improving the inference quality of the TKGR model. In addition, to better validate the effectiveness of the model, this paper constructs three new datasets to measure the inference accuracy, generalization, and robustness of the TKGR model in transitive, inductive, and few shot scenarios.

**Strengths:**

1.Solid theoretical analysis: The descriptions of constructing the temporal relation structure graph and reasoning path search algorithm are relatively clear.

2.Adequate experimental validation: The effectiveness of the model in reasoning accuracy and generalization performance was verified through transitive, inductive, and few shot scenarios.

3.The proposed new datasets expand the evaluation benchmarks in the TKGR field.

**Weaknesses:**

1.The description of challenge in the abstract and introduction does not correspond well. “Nonetheless, the majority of paths between the subject and object......lacking direct connectivity between subject and object entities” in the introduction seems to focus more on discussing the first challenge in the abstract, and the "Insufficient utilization of structural and textual information" in the abstract does not provide corresponding motivation and background analysis in the introduction.

2.The joint encoding method of time and text sequences seems a bit outdated. Although the experimental results demonstrate the effectiveness of this method, I would like to know if utilizing some more cutting-edge LLM based TKG learning models (e.g., [1], [2], [3]) can further improve the model's performance.

[1] Wang, Jiapu, et al. "Large Language Models-guided Dynamic Adaptation for Temporal Knowledge Graph Reasoning." arXiv preprint arXiv:2405.14170 (2024).

[2] Xia, Yuwei, et al. "Enhancing temporal knowledge graph forecasting with large language models via chain-of-history reasoning." arXiv preprint arXiv:2402.14382 (2024).

[3] Luo, Ruilin, et al. "Chain of history: Learning and forecasting with llms for temporal knowledge graph completion." arXiv preprint arXiv:2401.06072 (2024).

3.Omission in the experiment sections: (i)The window parameter w for the YAGO dataset was not provided. (ii)The value of δ in Formula 3 is not explicitly given. (iii)It seems that Figure 5 in the appendix is not described in the text and has no clear indication of the experimental dataset.

4. The quality of the presentation is below ICLR 2025 standards. For example, the format of the references should be consistent to ensure neatness and professionalism. For instance, the names of conferences should be uniformly presented either in abbreviations or full names, rather than a mixture of both.

**Questions:**

Besides the issues in weakness, there are a few other issues I would like to know:

1.For the PLM module in the joint coding model, does the authors verify the adaptability of the proposed model to different PLMs.

2.I would like to know if it is possible to consider training the model on a single dataset and evaluating it on different test sets in the context of inductive learning. For example, after training on ICEWS14, the model's performance on ICEWS18 and ICEWS05-15 can be on par with SOTA's baseline.

---

> ### Author Response · Authors · 2024-11-24
> **Response to Reveiewer pLWV Part 1**
>
> Thank you so much for the thoughtful questions and suggestions. We hope that our response below will address your concerns.
> >**W1**: The description of challenge in the abstract and introduction does not correspond well.
>
> Thank you for your valuable feedback. In the revised version, we have added a motivation and background analysis for Limitation 2: the insufficient utilization of structural and textual information, in the abstract and introduction sections. For your convenience, the additional content is presented as follows:
>
> "Furthermore, these methods insufficiently utilize the rich structural and textual information in temporal graphs. Symbolic methods approaches leverage the frequency information of relations, while neural methods focus on graph structural information. Both overlook the positive role of the structural dependencies between relations and textual semantics in logical reasoning."
>
> >**W2**: The joint encoding method of time and text sequences seems a bit outdated. Although the experimental results demonstrate the effectiveness of this method, I would like to know if utilizing some more cutting-edge LLM based TKG learning models (e.g., [1], [2], [3]) can further improve the model's performance.
>
> Thanks for your suggestions. References [1-2] have attempted to integrate LLMs to enhance reasoning on temporal knowledge graphs (TKG). They focus on utilizing chain-of-thought prompting templates to guide LLMs in inferring future events from historical event chains. However, these methods are challenging for untrained LLMs, as they can only perform semantic reasoning based on limited context without explicitly leveraging temporal patterns and the underlying logical correlations between events.
>
> In future work, we plan to explicitly integrate temporal logical rules with the strong reasoning capabilities of LLMs. Although Reference [3] also utilizes logical rules and LLMs, the LLMs are only involved in rule generation. In contrast, we aim to incorporate LLMs into the reasoning phase. Additionally, LLMs can be employed to interpret textual descriptions, assisting pre-trained language models in understanding textual semantics.
>
> >**W3**: Omission in the experiment sections: (i)The window parameter w for the YAGO dataset was not provided. (ii)The value of δ in Formula 3 is not explicitly given. (iii)It seems that Figure 5 in the appendix is not described in the text and has no clear indication of the experimental dataset.
>
> Thank you for your valuable feedback. We sincerely apologize for any confusion caused by our oversights. In response to your concerns, we have supplemented the experimental section in the revised version to avoid misunderstandings caused by missing details. These supplements include: (i) clarifying the setting of the time window parameter $\omega$ for the YAGO dataset, (ii) providing the settings for the hyperparameter $\delta$ and the margin parameter of confidence $\gamma$, and (iii) adding a description in the appendix regarding Figure 5, including details about the experimental dataset and result analysis. Please refer to Appendices C.2 and C.5 in the revised version for the aforementioned modifications.
>
> >**W4**: The quality of the presentation is below ICLR 2025 standards. For example, the format of the references should be consistent to ensure neatness and professionalism. For instance, the names of conferences should be uniformly presented either in abbreviations or full names, rather than a mixture of both.
>
> Thanks for your valuable feedback. We sincerely apologize for the oversight in the formatting of our reference list. We have carefully revised and adjusted the references to ensure they are consistent and well-organized. Please refer to the References section in the revised version to review our updates. Additionally, we have thoroughly reviewed the presentation and structure of the paper to ensure it meets the standards of ICLR 2025.
>
> **Reference:**
>
> [1] Xia, Yuwei, et al. "Enhancing temporal knowledge graph forecasting with large language models via chain-of-history reasoning." arXiv preprint arXiv:2402.14382 (2024).
>
> [2] Luo, Ruilin, et al. "Chain of history: Learning and forecasting with llms for temporal knowledge graph completion." arXiv preprint arXiv:2401.06072 (2024).
>
> [3] Wang, Jiapu, et al. "Large Language Models-guided Dynamic Adaptation for Temporal Knowledge Graph Reasoning." arXiv preprint arXiv:2405.14170 (2024).

---

> > ### Author Response · Authors · 2024-11-24
> > **Response to Reviewer pLWV Part 2**
> >
> > >**Q1**: For the PLM module in the joint coding model, does the authors verify the adaptability of the proposed model to different PLMs.
> >
> > Our CoLR demonstrates strong adaptability to different PLMs. In the revised manuscript, we conducted experiments on ICEWS14 using BERT, ALBERT, and MPNet as the backbone PLMs for CoLR. The results are showed in Figure 6(a). It can be observed that there are only minor performance differences among the three models, which may be attributed to variations in their sentence understanding capabilities. The differences can be further narrowed through fine-tuning. Overall, our CoLR is insensitive to the choice of PLM, exhibiting excellent flexibility.
> >
> > >**Q2**: I would like to know if it is possible to consider training the model on a single dataset and evaluating it on different test sets in the context of inductive learning. For example, after training on ICEWS14, the model's performance on ICEWS18 and ICEWS05-15 can be on par with SOTA's baseline.
> >
> > We believe that our proposed CoLR can address the hypothesis you raised due to its strong inductive capabilities. First, CoLR captures the cohesion between relations by constructing a temporal relation structure graph. As shown in Figure 4, the structural dependency between relations remains stable across different datasets. Since CoLR makes predictions based on the logical connections between relations, which are entity-agnostic, it maintains consistent reasoning performance even when the entity sets differ. Furthermore, when new relations appear in a dataset, CoLR leverages the PLM to capture the logical associations between relations from textual semantics. As a result, CoLR can be trained on one dataset and applied to reasoning tasks on multiple datasets while maintaining stable performance. As shown in Table 5 in Appendix C.4, despite being trained on ICEWS14, CoLR outperforms the SOTA baseline ILR-IR on ICEWS18 and ICEWS05-15.
> >
> > Thank you again for your feedback and suggestions. We hope that our thorough responses, along with the new results, will further underscore the value of our work. Your insights are invaluable in refining our paper. Please let us know if you have any further questions or concerns. We are committed to improving our paper and value your feedback.

---

### Official Review · Reviewer_otAn · 2024-11-03

**Soundness:** 2
**Presentation:** 2
**Contribution:** 2
**Rating:** 5
**Confidence:** 4

**Summary:**

The authors primarily address the shortcomings of rule-based reasoning methods in the field of temporal knowledge graph extrapolation. To this end, they propose two novel graph structures: the Time-Fusion Search Graph (TFSG) and the Temporal Relation Structure Graph (TRSG). Furthermore, they introduce the CoLR model, which employs a two-phase framework to mine relational dependencies and semantic structures within temporal knowledge graphs. The experimental results demonstrate the effectiveness of the proposed methods, validating their ability to make predictions and capture structural information in sparse data scenarios.

**Strengths:**

1. Innovatively Proposed the Temporal Relation Structure Graph (TRSG) Structure，which effectively captures stable structural information in temporal graphs.
2. Treated Rules as Text Sequences. By utilizing pre-trained text sequence encoders and time series encoders for learning, neural-symbolic integrated reasoning is achieved.
3. In the paper, three novel datasets are introduced, accompanied by comprehensive experiments designed to rigorously validate the performance of the proposed methods from multiple perspectives.

**Weaknesses:**

There are some issues in the main experimental results section.

1. The tasks in the field of temporal knowledge graph extrapolation vary. For instance, CENET uses triplet filtering for results and performs future predictions at any time point, whereas TiRGN employs quadruplet filtering for results and only predicts queries for the next time point. The authors' direct comparison of these two methods is obviously unreasonable.

2. Additionally, the paper does not specify how the baseline results were obtained, and the provided code link is inaccessible. The authors need to specify the type of tasks performed for the results obtained using the CoLR model, as well as whether triplet or quadruplet filtering was applied to these results.

**Questions:**

See the Weaknesses.

---

> ### Author Response · Authors · 2024-11-24
> **Responses to Reviewer otAn**
>
> Thank you for investing your time and expertise in reviewing our work. We are grateful for your recognition of our contributions in conceptual innovation, theoretical justification, reproducible, and benchmark construction, and we are delighted to clarify the concerns and answer the questions you raised.
>
> >**Q1**:The tasks in the field of temporal knowledge graph extrapolation vary. For instance, CENET uses triplet filtering for results and performs future predictions at any time point, whereas TiRGN employs quadruplet filtering for results and only predicts queries for the next time point. The authors' direct comparison of these two methods is obviously unreasonable.
>
> Thank you for your insightful comments regarding the comparison results. Upon careful examination, we found that the experimental setup of CENET was indeed inconsistent with other baselines, including our CoLR. To ensure a fairer comparison, we followed the experimental setup of TiRGN and re-reported CENET's results under the time-aware filtering setting. The updated experimental results can be found in Table 1. Notably, under a unified experimental setup, our CoLR achieves the best performance across all datasets, significantly outperforming the second-best baseline.
>
> >**Q2**: The paper does not specify how the baseline results were obtained.
>
> The experimental results of all baselines on the existing benchmark datasets are taken from the highest reported results in previous papers. Considering that CENET’s experimental setup differs from other baselines, we reproduced its results on the ICEWS and YAGO datasets under the time-filtering setting for a fair comparison. For the proposed dataset, we conducted experiments for each baseline, including our CoLR, using their parameter settings from ICEWS14 and reported the results. In the revised manuscript, we have provided details about how we obtained the baseline results in line 399-408 of the experiments section.
>
> >**Q3**: The provided code link is inaccessible.
>
> Thank you for your attention to our work. In the initial submitted version, the core code of our method has been open-sourced through the [anonymous code link](https://anonymous.4open.science/r/CoLR-0839) provided in the paper. We will release all our code, including the newly proposed datasets, immediately after the paper is accepted.
>
> >**Q4**: The authors need to specify the type of tasks performed for the results obtained using the CoLR model, as well as whether triplet or quadruplet filtering was applied to these results.
>
> Thank you for your insight feedback. Following the TiRGN setup, we performed quadruplet filtering under the time-filtering setting and only predicted future events at the next timestamp. We have clarified our task type in Appendix C.2 of the revised version.
>
> Thank you once again for your valuable feedback and comments! If there are any further questions or aspects you feel remain unaddressed, we are more than willing to provide additional information and clarifications as needed.

---

> ### Comment · Reviewer_otAn · 2024-11-26
>
> The ICEWS series of datasets may face data leakage issues due to the use of external knowledge. Meanwhile, the newly constructed ACLED2023 dataset is too small and singular, making it insufficient to demonstrate the effectiveness of the method.

---

> > ### Author Response · Authors · 2024-11-27
> >
> > Thank you for your response and feedback! In response to the questions you raised regarding the dataset, we provide the following clarifications to address your concerns.
> > >The ICEWS series of datasets may face data leakage issues due to the use of external knowledge.
> >
> > We chose the ICEWS series datasets as they are among the most widely used benchmark datasets for temporal knowledge graph reasoning. As shown in Table 1 in the manuscript, rule-based methods only reported results on ICEWS datasets. Therefore, experiments on ICEWS provide a more direct demonstration of our method's superior performance.
> >
> > We acknowledge your concern regarding potential data leakage in the ICEWS series datasets due to our incorporation of pre-trained language models (PLMs). To address this, we have also reported our CoLR's experimental results on YAGO, another widely-used benchmark dataset, where CoLR consistently outperforms other baselines, validating the effectiveness of our proposed method. Moreover, compared to PPT, which also incorporates PLMs, our method achieves nearly double the performance improvement, suggesting that potential data leakage risks are either non-existent or minimal for PLMs. We present the comparison with PPT's on the MRR metrics as follows:
> > | Method | ICEWS14 | ICEWS18 | ICEWS05-15 |
> > |--------|---------|---------|------------|
> > | RE-GCN | 37.78   | 27.51   | 38.27      |
> > | PPT    | 38.24   | 26.63   | 38.85      |
> > | CoLR   | 75.72   | 68.74   | 76.82      |
> >
> > As shown in the table, PPT, despite incorporating BERT, performs worse than the traditional baseline RE-GCN on the ICEWS18 dataset, further confirming that the impact of potential data leakage of ICEWS datasets is negligible. Additionally, to explicitly prevent data leakage risks, we constructed the ACLED2023 dataset. As described in Section 6.1, lines 353-358 of the manuscript, ACLED2023 effectively avoids data leakage by introducing events that PLMs could not have been exposed to. The experimental results on ACLED2023 demonstrate that CoLR's performance stems from the proposed method itself rather than data leakage.
> >
> > >Meanwhile, the newly constructed ACLED2023 dataset is too small and singular, making it insufficient to demonstrate the effectiveness of the method.
> >
> > With the recent trend of leveraging PLMs and LLMs for temporal knowledge graph reasoning tasks, the knowledge contained in existing benchmark datasets may have already been learned by these language models as part of their training corpus. To mitigate the risk of data leakage affecting model performance, we constructed the ACLED2023 dataset. As shown in Table 1 of the manuscript, CoLR consistently outperforms existing baselines on this dataset, demonstrating that its superior performance is unrelated to data leakage. This also ensures the validity of CoLR's results on other datasets, which have not benefited from potential leakage. Furthermore, as illustrated in Table 4 of Appendix C.3, the scale of ACLED2023 is comparable to the conventional benchmark dataset ICEWS14. Therefore, we believe that the experimental results on ACLED2023 sufficiently demonstrate the effectiveness of CoLR. Based on your suggestion, we plan to consider expanding ACLED2023 into ACLED2024 in future work to provide a more robust evaluation of model performance.
> >
> > Thank you again for your response. If you have any further questions or suggestions, please do not hesitate to let us know.

---

### Official Review · Reviewer_R1ug · 2024-11-04

**Soundness:** 3
**Presentation:** 3
**Contribution:** 3
**Rating:** 6
**Confidence:** 3

**Summary:**

This paper proposes a two-stage framework named CoLR for coherent logical reasoning over TKGs. The framework integrates relation dependencies and textual semantics to enhance the performance of link forecasting tasks. The key contributions include the construction of a temporal relation structure graph (TRSG) to capture structural dependencies, a time-fusion search graph (TFSG) to efficiently extract reliable temporal paths, and the encoding of textual and timestamp sequences using pre-trained language models and time sequence encoders. The authors construct three new datasets to comprehensively evaluate the model, demonstrating SOTA performance across transductive, inductive, and few-shot scenarios.

**Strengths:**

1. The approach of combining relation dependencies and textual semantics in a two-stage framework for TKG reasoning is novel. The TRSG and TFSG concepts, along with the path supplement strategy, are innovative contributions.
2. The methodology is rigorously designed and theoretically grounded. The proof for the cohesion matrix calculation and the algorithm for path extraction are well-documented.
3. The paper is well-structured and clearly written. Definitions, formulations, and algorithms are explained in detail, making the approach reproducible.
4. The proposed model achieves state-of-the-art results on multiple datasets, demonstrating its effectiveness and generalizability. The new datasets provide valuable resources for future research in this domain.

**Weaknesses:**

While the scalability of the approach is discussed, concrete experiments demonstrating its performance on larger TKGs are missing. The computational complexity of the TRSG construction and path extraction could become a bottleneck for very large graphs.
The authors could consider including visualizations of the TRSG and TFSG to intuitively illustrate their structures and how they facilitate path extraction.

**Questions:**

How sensitive is the model to the choice of pre-trained language model? Have you experimented with different language models to see the impact on performance?

---

> ### Author Response · Authors · 2024-11-24
> **Responses to Reviewer R1ug**
>
> We appreciate your thorough review as well as constructive feedback, and we try to answer your concerns and questions as follows.
> >**W1**: While the scalability of the approach is discussed, concrete experiments demonstrating its performance on larger TKGs are missing. The computational complexity of the TRSG construction and path extraction could become a bottleneck for very large graphs.
>
> Thank you very much for your valuable feedback. In the revised version, we have added efficiency analyses of TRSG and path search in Appendices A.3 and C.5, respectively.
>
> For TRSG construction, larger TKGs do not significantly lead to computational bottlenecks. As shown in Equation 2, the TRSG on each subgraph can be obtained by multiplying two entity-relation matrices, which is efficiently implemented in PyTorch. Therefore, as the graph size expands, the computational cost mainly arises from the increased number of matrix multiplications due to the extension of the timestamp sequence. For a TKG with $N$ timestamps, when the time window $\omega$ is 1, the times of matrix multiplications is $N$; when the time window size is $N$, the computation times is $1/2(N*(N+1))$. Thus, the time complexity for constructing a TRSG lies between $O(N)$ and $O(N^2)$. Since the time window is typically much smaller than $N$, the time complexity of constructing TRSG approaches $O(N)$. Clearly, as the number of timestamps in the TKG increases, the computational cost does not grow dramatically. For example, in the ICEWS05-15 dataset with over 4000 timestamps, constructing a TRSG with a time window of 10 takes only 20 seconds. ICEWS05-15 is the second-largest TKG benchmark dataset in terms of the number of timestamps, surpassed only by GDELT. However, we did not conduct experiments on GDELT because the textual descriptions of relations in GDELT are presented in a special coded format. Nonetheless, we believe the experimental results on ICEWS05-15 sufficiently demonstrate that our method can effectively scale to larger datasets.
>
> For path extraction, by introducing TRSG and TFSG, our time-fusion path search algorithm can efficiently identify historical paths relevant to the query. Specifically, TRSG ensures that CoLR only needs to retrieve the top-K historical paths, avoiding the additional computational cost of retrieving unrelated paths. TFSG preserves temporal information while compressing the TKG into a static graph, eliminating the need for re-expanding the static graph back into a TKG. Similarly, taking ICEWS05-15 as an example, CoLR can complete the historical path search for all quadruplets within three minutes, while TLogic requires approximately 15 minutes. The detailed experimental results are shown in Table 8 of the revised version.
>
> In summary, we believe that the computational complexity of TRSG construction and path retrieval will not become a bottleneck for applying CoLR to large-scale graphs.
>
> >**W2**: The authors could consider including visualizations of the TRSG and TFSG to intuitively illustrate their structures and how they facilitate path extraction.
>
> Thank you very much for your suggestions. In the initial submitted version, we have done the visualizations of the TRSG and TFSG in Figures 2(d) and 3(b), respectively. To further clarify their structures and their roles in path extraction, we plan to release a demonstration video of graph construction and path search on the paper's homepage after its publication.
>
> >**Q1**: How sensitive is the model to the choice of pre-trained language model? Have you experimented with different language models to see the impact on performance?
>
> Our CoLR is insensitive to the choice of pre-trained language models. In the revised manuscript, we provided sensitivity experiments regarding CoLR's performance with different PLMs. As shown in Figure 6(a), replacing the PLM only caused minor performance differences, which can be attributed to the varying sentence understanding capabilities of different language models. The differences can be further narrowed through fine-tuning. Therefore, our CoLR is insensitive to the choice of PLM, exhibiting excellent flexibility.
>
> Thank you once again for your valuable feedback and comments! If there are any further questions or aspects you feel remain unaddressed, we are more than willing to provide additional information and clarifications as needed.

---

> > ### Comment · Reviewer_R1ug · 2024-12-03
> > **Thanks**
> >
> > Thank you for addressing my concerns! I think I will keep my rating.

---

### Official Review · Reviewer_aa3E · 2024-11-05

**Soundness:** 3
**Presentation:** 3
**Contribution:** 3
**Rating:** 6
**Confidence:** 2

**Summary:**

The paper introduces CoLR, a two-stage framework designed to improve logical reasoning over TKGs by integrating structural dependencies and textual semantics. The approach constructs a Temporal Relation Structure Graph to identify relations and their temporal cohesion, alongside a Time-Fusion Search Graph for reliable path searching. CoLR then encodes both the textual content and timestamp sequences with a pre-trained language model and a time sequence encoder, enhancing predictive reasoning on TKGs. Experiment results shows that CoLR significantly outperforms previous methods.

**Strengths:**

1.	The CoLR framework effectively combines structural dependencies and textual semantics, which improves logical reasoning over TKGs.
2.	By constructing new datasets tailored for specific reasoning challenges, the authors ensure a thorough evaluation of their model's capabilities in transductive, inductive, and few-shot scenarios.
3.	The TRSG and TFSG facilitate efficient pathfinding and extraction, reducing computational costs and enabling the model to handle complex reasoning tasks.

**Weaknesses:**

1.	The paper does not provide a detailed analysis of CoLR's computational complexity. Metrics such as runtime comparisons are not thoroughly discussed. The time-fusion path extracting process, especially with the addition of the TRSG and TFSG, might introduce computational bottlenecks. It's unclear how the proposed method scales with larger graphs or longer time windows.
2.	While an ablation study is presented, it might not comprehensively cover all components of the model. For example, the impact of different path lengths (L) or the number of paths (K) on performance is not explored.
3.	The framework’s reliance on cohesive relations may lead to suboptimal explanations for events connected by low-cohesion paths, affecting interpretability in sparse data scenarios.

**Questions:**

None

---

> ### Author Response · Authors · 2024-11-24
> **Responses to Reviewer aa3E**
>
> Thank you for your valuable feedback. We are greatly delighted to note your recognition of the contributions our paper makes in terms of method design, concept proposal, and benchmark construction. We are happy to address the questions you’ve raised.
> >**W1**: The paper does not provide a detailed analysis of CoLR's computational complexity. Metrics such as runtime comparisons are not thoroughly discussed.
>
> In the revised manuscript, we have provided an analysis of computational complexity and a comparison of runtime with other methods in Appendix A.3 and Appendix C.5, respectively. The primary time cost of our CoLR comes from the fine-tuning of the PLM, which is the main reason our method requires longer training time compared to other baselines. Despite our adoption of LoRA to enhance training efficiency, the training still consumes several hours. However, when using GRU as the encoder, CoLR-GRU achieves comparable training and inference efficiency with other baselines while maintaining optimal performance. Moreover, a more advantageous approach would be to use the PLM solely as an encoder to initialize entity and relation embeddings without participating in forward and backward propagation. It allows us to leverage the prior semantics of the PLM while ensuring efficient training and inference.
>
> >**W2**: The time-fusion path extracting process, especially with the addition of the TRSG and TFSG, might introduce computational bottlenecks. It's unclear how the proposed method scales with larger graphs or longer time windows.
>
> We completely understand your concerns; however, we believe that the introduction of TRSG and TFSG will not cause a computational bottleneck for path extraction. On the contrary, TRSG and TFSG are key to improving the efficiency of path search. Specifically, TRSG ensures that CoLR only needs to retrieve the top-K historical paths, avoiding the additional computational cost of retrieving unrelated paths. TFSG preserves temporal information while compressing the TKG into a static graph, eliminating the need for re-expanding the static graph back into a TKG. Since we only need to retrieve the top-K paths most relevant to the query, the time required for path retrieval does not increase linearly with the graph size or time window. In Table 8 of the revised version, we provide comparative experiments on path search efficiency.CoLR completes historical path retrieval for all quadruplets on the ICEWS14 dataset in 40 seconds, while on ICEWS05-15, which has ten times the timestamps of ICEWS14, it only takes 3 minutes. In comparison, TLogic requires 15 minutes to complete path retrieval on ICEWS05-15.
>
> >**W3**: While an ablation study is presented, it might not comprehensively cover all components of the model. For example, the impact of different path lengths (L) or the number of paths (K) on performance is not explored.
>
> Thank you very much for your professional comments. In the revised manuscript, We have provided a more comprehensive evaluation and analysis of our CoLR in Appendix C.5, including scalability assessments and sensitivity evaluations regarding time window $\omega$ and the number of paths. Based on your suggestions and those of other reviewers, we have added sensitivity analyses for the pre-trained encoder and the maximum path length along with efficiency analyses on training time and path search time. Please refer to Appendix C.5 in the revised version for detailed experimental results and analysis.
>
> >**W4**: The framework’s reliance on cohesive relations may lead to suboptimal explanations for events connected by low-cohesion paths, affecting interpretability in sparse data scenarios.
>
> We fully understand your concerns. Similar to other methods that rely solely on frequency information for path extraction, if CoLR depends only on the cohesion between relations, it may fail to explain paths with low cohesion. Therefore, we consider incorporating additional information during path extraction, such as relation frequency. Moreover, these paths can also be interpreted through PLM by leveraging textual semantic relevance. For instance, two relations that are disconnected on the graph may still be logically related in terms of textual semantics.
>
> Thank you again for reviewing our paper and for the pleased comments. We hope that our response and clarification have addressed your questions and concerns. We sincerely invite you to engage with us if you have more questions.

---

### Meta-Review · Area_Chair_At2d · 2024-12-22

**Metareview:**

(a) Scientific Claims and Findings

The paper introduces a novel method, CoLR, that integrates structural and textual semantics for reasoning over temporal knowledge graphs (TKGs). Key innovations include the TRSG for relation cohesion analysis and TFSG for efficient temporal path searching. Results on existing benchmarks and newly proposed datasets show state-of-the-art performance across transductive, inductive, and few-shot scenarios.

(b) Strengths
- Novel combination of structural dependencies and textual semantics in TKG reasoning.
- Rigorous theoretical grounding and method design, including cohesion matrix computation and efficient path extraction.
- Introduction of three datasets tailored for specific reasoning challenges.

(c) Weaknesses
- Limited scalability analysis; concerns about computational complexity were not fully addressed initially.
- Some experimental comparisons (e.g., CENET) were based on inconsistent setups, later corrected in the rebuttal.
- Concerns regarding the adaptability of methods to newer large language models.
- Presentation issues in earlier submissions, including inconsistent formatting and missing details in experimental descriptions.

(d) Decision: reject

While the paper proposes an interesting framework, unresolved concerns about scalability, reliance on older pre-trained models, and limited dataset scope undermine its contributions. Presentation issues and inconsistent experimental setups further detract from its quality. Importantly, the reviewers did not strongly support the acceptance of the paper, and two of the favorable scores both come with a low confidence level, leading to insufficient support for acceptance.

**Additional Comments On Reviewer Discussion:**

During the rebuttal phase, the authors addressed reviewer concerns effectively:

- Reviewer aa3E: Concerns about computational complexity and ablation studies were addressed with added sensitivity analyses and runtime comparisons, showing efficiency improvements with TRSG/TFSG.
- Reviewer R1ug: Scalability concerns and adaptability to different pre-trained models were clarified, with added experiments demonstrating minor performance variation across models.
- Reviewer otAn: Issues with experimental setups (e.g., CENET comparisons) were corrected, and data leakage concerns in ICEWS datasets were mitigated by additional experiments on YAGO and ACLED2023.
- Reviewer pLWV: Presentation issues were resolved, and omitted details were clarified. The authors also highlighted the model's inductive capabilities, demonstrating stable performance across datasets in transfer learning scenarios.

Despite these efforts, two reviewers (otAn, pLWV) maintained slightly below-threshold scores due to lingering concerns about dataset limitations and the use of older modeling techniques.

---

### Decision · Program_Chairs · 2025-01-22

Reject